# DMM: Distributed Matrix Mechanism for Differentially-Private Federated Learning Based on Constant-Overhead Linear Secret Resharing

Alexander Bienstock [1]  Ujjwal Kumar [2]  Antigoni Polychroniadou [1]

## Abstract

Federated Learning (FL) solutions with central Differential Privacy (DP) have seen large improvements in their utility in recent years arising from the *matrix mechanism*, while FL solutions with distributed (more private) DP have lagged behind. In this work, we introduce the *distributed* matrix mechanism to achieve the best-of-both-worlds; better privacy of distributed DP and better utility from the matrix mechanism. We accomplish this using a novel cryptographic protocol that securely transfers sensitive values across client committees of different training iterations with constant communication overhead. This protocol accommodates the dynamic participation of users required by FL, including those that may drop out from the computation. We provide experiments which show that our mechanism indeed significantly improves the utility of FL models compared to previous distributed DP mechanisms, with little added overhead.

## 1. Introduction

In Federated Learning (FL), a machine learning model is trained using data from several end-users/clients. Since such data can often be sensitive, a key challenge in FL is maintaining utility of the trained models, while preserving privacy of the end-users. FL has experienced an explosion of progress in recent years, both in industry and research.

In more detail, in each training iteration of FL, typically a central server sends the current model parameters to a set of clients, which we call a *committee*, who locally execute a step of Stochastic Gradient Descent on their own data to obtain gradients with respect to a loss function. These

gradients are then aggregated and sent to the central server using different techniques to update the model parameters for the next iteration (e.g., (McMahan et al., 2016; Fallah et al., 2020; Sahu et al., 2018)).

The main privacy metric for FL is differential privacy (DP) (Dwork et al., 2006). Roughly speaking, DP guarantees that with high probability, one cannot tell whether a user participated in a given FL execution. There are two different notions of DP that can be considered. In *central* DP, there is a centralized server who receives the aggregated gradients from the clients in each iteration (perhaps using a Secure Aggregation protocol (Kairouz et al., 2021a; Bonawitz et al., 2017; Liu et al., 2022; Karthikeyan & Polychroniadou, 2024; Li et al., 2023)) and then updates the model by adding its own DP noise to these aggregated gradients. See the left side of Figure 1 for a flowchart illustrating the process. In this case, DP holds with respect to those to whom the server sends the updated models (assuming that the server did indeed add noise), but not the server itself. In *distributed* DP, there may still be a centralized server, however, the clients utilize a Secure Aggregation protocol to release to the server *only* an aggregation of their gradients with their own DP noise already added in. Thus, DP holds with respect to the server as well; in particular, the clients do not need to trust the server to add noise. Indeed, distributed DP is important for particularly sensitive data that cannot be known by *anyone* else and for which we cannot rely on a central server to protect.

There has been tremendous progress recently in the area of central DP for FL, e.g., (Choquette-Choo et al., 2023a; Dvijotham et al., 2024; McMahan et al., 2024). These works use a sophisticated set of techniques from the DP literature called the *matrix mechanism* (Hubert Chan et al., 2010; Dwork et al., 2010) to achieve excellent privacy-utility trade-offs. Indeed, in this setting, since the central server receives all of the gradients in the clear and samples all noise on its own, it can *correlate* the noise across iterations in a complex manner. Intuitively, this means that noise can be re-used across iterations so that the cumulative noise across all iterations is lower compared to sampling new, fresh noise to hide the gradients in each iteration.

On the other hand, in the setting of distributed DP, the

[1]J.P. Morgan AI Research & J.P. Morgan AlgoCRYPT CoE, New York, New York, USA [2]J.P. Morgan, Mumbai, India. Correspondence to: Alexander Bienstock <alex.bienstock@jpmchase.com>.

*Proceedings of the 42$^{nd}$ International Conference on Machine Learning*, Vancouver, Canada. PMLR 267, 2025. Copyright 2025 by the author(s).

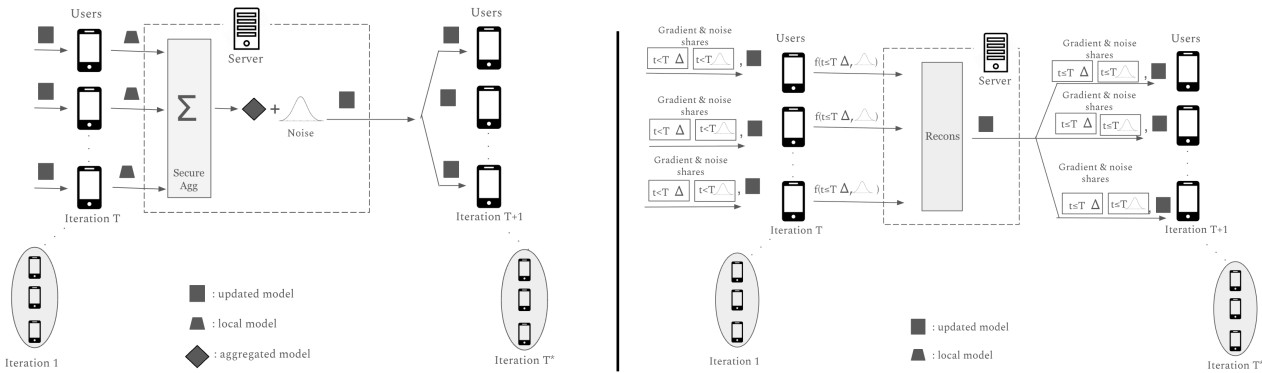

*Figure 1.* **Left**: FL in the central DP model. Users in iteration $T$ update the model locally and these updates are aggregated to the server. The server then adds noise itself before sending the updated model to committee $T + 1$. **Right**: FL based on DMM in the distributed DP model. Users in iteration $T$ receive noise and gradient shares from previous iterations. These parties combine the received shares with shares of their new gradients and freshly sampled noise via a linear combination, $f$, and send these combined shares to the server who uses them to reconstruct (only) the updated model. Afterward, the gradient and noise shares are reshared to the parties in committee $T + 1$, ensuring continuity in DMM.

clients just add noise locally to their gradients (Kairouz et al., 2021a; Agarwal et al., 2021; Chen et al., 2022). Since clients change in each iteration, the noise cannot be correlated across epochs via the matrix mechanism like in the central DP setting, and so the privacy-utility trade-off of distributed DP pales in comparison to that of central DP thus far.

**Our Contributions.** In this work, we propose a solution to achieve the "best-of-both-worlds" of the central and distributed DP settings, called the *Distributed Matrix Mechanism* (DMM). We achieve privacy against the central server, i.e., distributed DP, while using correlated noise to get privacy-utility trade-offs matching the central DP setting.[1]

**1)** DMM starts with *linear secret sharing* (Shamir, 1979), a central technique in the cryptographic literature which has also been used for FL, e.g., (Ma et al., 2023; Shao et al., 2022; Marchand et al., 2022). Secret sharing allows for a *dealer* party to distribute to $n$ parties different *shares* of some secret $x$, such that any $t_c$ (corrupted) parties cannot learn anything about $x$ from all of their shares, while any $t_c + k$ parties, for some $k > 0$ can use their shares to *reconstruct* $x$. These shares are also *linear*, meaning that if the users have shares of $x_1$ and $x_2$, they can add their shares together to obtain a sharing of $x_3 = x_1 + x_2$.

Typically in FL, secret sharing is used by a single set of parties to secret share (noisy) data to the other parties in the set. In our setting, however, we additionally need the noise and gradients from users in a given committee to somehow be *reshared* to users in future committees. Thus, we develop new techniques in this paper to build our *constant-overhead*

*linear secret resharing protocol*, LRP. Indeed, our new techniques are paramount, since the naive way to perform such resharing costs $n^2$ communication per secret, instead of $O(1)$ per secret, where $n$ is the number of parties in each committee. We can see from Table 1 that this results in communication as low as 25.1 MB per client using our new techniques, and infeasible communication as high as 2.13 TB per client using the naive resharing. See Section 3 for details on the naive secret resharing protocol and our LRP protocol with constant communication overhead.

**2)** Given LRP, we can instantiate the matrix mechanism in a distributed fashion to obtain DMM: First, the parties take linear combinations of the secret shared gradients and noise, thus introducing noise correlations across epochs. Then, the parties can reconstruct these aggregated gradients with (correlated) noise to the server. Finally, users (re)share the gradients and noise using LRP. See the right side of Figure 1 for a flowchart illustrating our approach.

DMM is detailed in Section 4. Importantly, DMM maintains DP even in the presence of corrupted parties who might manipulate their shares of the gradients and noise. Moreover, DMM achieves *dropout tolerance*: In FL, the gradients from end-users often come from mobile devices, and therefore it may not be guaranteed that such users will stay online for the whole training iteration, even if they are honest. Thus, the protocol must not fail if some (honest) users drop out.

**3)** We implement the Distributed Matrix Mechanism using our resharing protocol and empirically test its efficacy in training differentially private FL models. For example, in Figure 2, we show that for Stack Overflow Next Word Prediction (Authors, 2019), our approach improves upon the privacy-utility tradeoff of the most accurate prior distributed DP approach, the Distributed Discrete Gaussian

---

[1]We note that, just as in (Kairouz et al., 2021a) and all other works using Secure Aggregation to obtain DP guarantees via aggregated noise, we actually obtain *computational* DP (Mironov et al., 2009).

|  | LRP Comp. | SecAgg Comp. | LRP Comm. | Naive SR Comm. | SecAgg Comm. |
|---|---|---|---|---|---|
| Opt. | 7.69 s | 61.3 ms | 4.68 GB | 2.13 TB | 16.2 MB |
| Hon. | 412 ms | 61.3 ms | 25.1 MB | 11.4 GB | 16.2 MB |

*Table 1.* Client computation and communication of our LRP resharing protocol, naive secret resharing, and SecAgg per training iteration on Stack Overflow Next Word Prediction for committee size $n = 64$. We give results for both the optimal (Choquette-Choo et al., 2023a) and more efficient Honaker online (Kairouz et al., 2021b; Honaker, 2015) matrix mechanisms. SecAgg is the bottleneck of prior distributed DP approaches (Kairouz et al., 2021a) and LRP (as opposed to naive secret resharing) is the bottleneck of DMM. (ms := milliseconds; s := seconds).

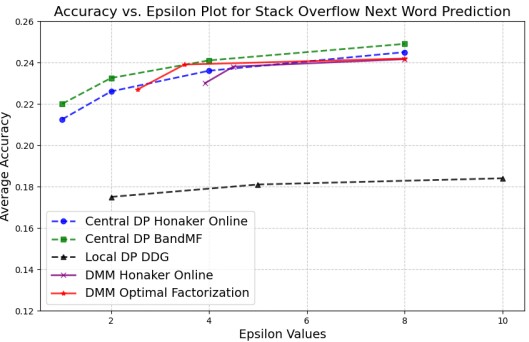

*Figure 2.* Test accuracies on Stack Overflow Next Word Prediction across different privacy levels $\varepsilon$ for the distributed DP DDG mechanism (Kairouz et al., 2021a), the central DP BandMF and Honaker online mechanisms (Choquette-Choo et al., 2023a), and our distributed DP DMM instantiated with the optimal (Choquette-Choo et al., 2023a) and Honaker online (Honaker, 2015) matrix factorizations. DMM performs 5-6 percentage points better than the prior distributed DP approach and similar (sometimes better) to the prior central DP approaches. We use $\delta = 1/N$ for $(\varepsilon, \delta)$-DP, where $N$ is the total number of clients selected across training.

(DDG) Mechanism (Kairouz et al., 2021a), and matches that of the best central DP approach (Choquette-Choo et al., 2023a). We show similar results in Figure 4 for Federated EMNIST (Caldas et al., 2018). It can also be observed in Tables 1 and 2 that our solution is lightweight. Indeed, DMM adds less than 10 seconds of computation and in some cases less than 2 MB of communication per client compared to the prior distributed DP approach of secure aggregation.

**Related Work.** In concurrent work, (Ball et al., 2024) take another approach to our problem, without secret sharing. They instead separately maintain aggregate noise and gradient encryptions across iterations using (linearly homomorphic) encryption. These encryptions are then added together by the server and decrypted using clients' noisy secret keys (that do not reveal the actual secret keys) in a clever fasion to reveal *only* the sums with correlated noise in each iteration. They also sketch a solution with dropout resilience. Ball et al. do not provide any code or straightforward method for calculating communication costs; however, we expect their communication complexity to be better than ours. Yet, since they rely on computational assumptions for linearly homomorphic encryption, whereas we just use information-theoretic secret sharing techinques, we expect ours to be faster. Moreover, Ball et al. do not have a maliciously secure protocol.

A fruitful line of works has used correlated noise, and in particular, the matrix mechanism to improve the privacy-utility tradeoff and memory costs of (central) DP FL for increasingly realistic multi-participation settings (Kairouz et al., 2021b; Denisov et al., 2023; Choquette-Choo et al., 2023b;a; Dvijotham et al., 2024; McMahan et al., 2024). Privacy amplification techniques like shuffling (Erlingsson et al., 2019; Feldman et al., 2022) or (Poisson) subsampling (Abadi et al., 2016; Zhu & Wang, 2019; Wang et al., 2019) are sometimes used to increase privacy-utility tradeoffs; however, these require strong assumptions on how data is processed which are often not suitable for FL in practice and thus should be avoided (Kairouz et al., 2021b).

New distributed DP mechanisms for FL, the Skellam Mechanism (Agarwal et al., 2021), and Distributed Mean Estimation (DME), the Poisson Binomial Mechanism (PBM) (Chen et al., 2022), have appeared in the literature recently, mostly improving the efficiency of DDG. Indeed, in these works, it is shown that roughly the same privacy-utility tradeoff as DDG is acheived by the Skellam Mechanism, while PBM is not compared empirically to DDG, nor are FL experiments with PBM provided (though they state that their asymptotic error for DME is the same as DDG). We therefore refer to (Kairouz et al., 2021a) as the state-of-the-art for privacy-utility tradeoff. Furthermore, PBM specifically does not seem suited to our techniques, since it departs from the *additive noise* paradigm.

Several works have considered so-called *proactive secret sharing* (Ostrovsky & Yung, 1991; Baron et al., 2014; Maram et al., 2019). This setting is similar to ours in which secrets are reshared, however, there the users stay the same in each iteration; just the users that are corrupted changes in each iteration. Papers that study a similar model to ours exist, but for more general computations than the special case of aggregation and without a central server that minimizes interaction between clients, and thus are inefficient (Gentry et al., 2021; Bienstock et al., 2023; Choudhuri et al., 2021; Rachuri & Scholl, 2022; Bienstock et al., 2025).

## 2. Preliminaries

### 2.1. Differentially Private Federated Learning

In this section, we define some notions important to DPFL. Let $T^*$ be the number of training iterations, $n$ the number of parties in each committee, and $d$ model dimension.

**Adjacency and Participation Schemas.** DP requires a notion of adjacent datasets. Two data streams $\boldsymbol{X}$ and $\tilde{\boldsymbol{X}}$ are adjacent if the data associated with any single user is altered, in every iteration in which the user participates.[2] The pattern of when this user participates does not change in these two adjacent streams. A *participation schema* $\Phi$ contains all possible *participation patterns* $\phi \in \Phi$, with each $\phi \subseteq [T^*]$ indicating a set of iterations in which a single user participates. Let Nbrs be the set of all pairs of neighboring streams $\boldsymbol{X}$ and $\mathfrak{D} = \{\boldsymbol{X} - \tilde{\boldsymbol{X}} : (\boldsymbol{X}, \tilde{\boldsymbol{X}}) \in$ Nbrs$\}$ represent the set of all possible differences between neighboring $\boldsymbol{X}, \tilde{\boldsymbol{X}}$. We say a $\mathfrak{D}$ satisfies the participation schema $\Phi$ if the indices of all nonzero rows in each $\mathbb{R}^{T^* \times d}$ matrix $\boldsymbol{U} \in \mathfrak{D}$ are a subset of some $\phi \in \Phi$.

**Centralized DP Matrix Mechanism.** Let $\boldsymbol{A} \in \mathbb{R}^{T^* \times T^*}$ be an appropriate linear query workload (e.g., prefix sums) that is publicly known to all participants. Matrix mechanisms in the central DP setting use a factorization $\boldsymbol{A} = \boldsymbol{B}\boldsymbol{C}$ to privately estimate the quantity $\boldsymbol{A}\boldsymbol{X}$ as $\widehat{\boldsymbol{A}\boldsymbol{X}} = \boldsymbol{B}(\boldsymbol{C}\boldsymbol{X} + \boldsymbol{Z})$, where $\boldsymbol{Z}$ is sampled by the central server from some noise distribution.

Each entry of the vector $\widehat{\boldsymbol{A}\boldsymbol{X}}$ corresponds to a model iteration that is released. The matrix $\boldsymbol{A}$ is lower-diagonal, which means that the $T$-th entry of $\widehat{\boldsymbol{A}\boldsymbol{X}}$ only depends on the first $T$ entries of $\boldsymbol{X}$, for each dimension. Additionally, the $T$-th entry of $\widehat{\boldsymbol{A}\boldsymbol{X}}$ depends on the first $T$ entries of $\boldsymbol{Z}$, which means that the noise used in each released model iteration is *correlated*.

We now define the *sensitivity* of the central DP matrix mechanism for a particular participation schema $\Phi$ with set of neighboring streams Nbrs as $\text{sens}_\Phi(\boldsymbol{C}) = \sup_{(\boldsymbol{X}, \tilde{\boldsymbol{X}}) \in \text{Nbrs}} ||\boldsymbol{C}\boldsymbol{X} - \boldsymbol{C}\tilde{\boldsymbol{X}}||_F = \sup_{\boldsymbol{U} \in \mathfrak{D}} ||\boldsymbol{C}\boldsymbol{U}||_F$.[3] As in previous works, it is useful to analyze $\text{sens}_\Phi(\boldsymbol{C})$ when all of the contributions from users are clipped to $\ell_2$ norm at most $c = 1$, noting that the actual value of $\text{sens}_\Phi(\boldsymbol{C})$ scales with $c$ in general. In our work, however, it is useful to explicitly define the sensitivity for contributions of $\ell_2$ norm $c = 1$ as $\text{sens}_\Phi^1(\boldsymbol{C})$. The expected total squared error on $\boldsymbol{A}$ is typically given as $\mathcal{L}(\boldsymbol{B}, \boldsymbol{C}) = \text{sens}_\Phi(\boldsymbol{C})||\boldsymbol{B}||_F^2$ and the goal is to find a factorization that minimizes this loss.

---

[2]We study the more general user-level DP in this work, as opposed to example-level DP.

[3]$|| \cdot ||_F$ is the Frobenius norm.

### 2.2. Problem Statement and Security Model

For each iteration $T \in [T^*]$, we have a committee of (different) clients $\mathcal{C}_T$. The clients in this committee receive the current model parameters $\theta$ from the server and some values (secret shares) from the previous committee $\mathcal{C}_{T-1}$. Each client $P_{T,i}$ uses $\theta$ and their private data to obtain gradients $\boldsymbol{g}_{T,i}$ and also samples noise $\boldsymbol{z}_{T,i}$ from some distribution $\mathcal{D}$. These clients then interact with each other and the server with the goal of revealing *only* $\widehat{\boldsymbol{A}\boldsymbol{X}}_T = \boldsymbol{A}_{[T:,]}\boldsymbol{X} + \boldsymbol{B}_{[T:,]}\boldsymbol{Z}$ to the server, where each entry $\boldsymbol{X}_T = \sum_{i=1}^n \boldsymbol{g}_{T,i}$ and $\boldsymbol{Z}_T = \sum_{i=1}^n \boldsymbol{z}_{T,i}$ for $T \in [T^*]$. We allow $t_c$ clients per iteration as well as the server to be *corrupted* by an adversary $\mathcal{A}$; i.e., $\mathcal{A}$ can use the values sent to the corrupted parties to try to learn anything besides each $\widehat{\boldsymbol{A}\boldsymbol{X}}_T$. In fact, we handle *malicious* adversaries that can send arbitrary values to other parties. We allow such adversaries to change each $\widehat{\boldsymbol{A}\boldsymbol{X}}_T$ received by the server by some additive $\chi_T$ factors that are *independent* of $\boldsymbol{g}_{T,i}, \boldsymbol{z}_{T,i}$ for $T \in [T^*]$ of the other clients; thus preserving DP. We also allow $t_d$ honest clients per iteration to drop out; in this case the correct $\widehat{\boldsymbol{A}\boldsymbol{X}}_T$ should still be received by the server (with added $\chi_T$ defined by the adversary). We require $t_d + t_c < (1/2 - \mu)n$, for constant $0 < \mu < 1/2$, to guarantee security. In Section C, we formalize this model using a standard simulation-style definition (Goldreich, 2004) and show that such adversaries cannot learn anything besides $\widehat{\boldsymbol{A}\boldsymbol{X}}$.

### 2.3. (Packed) Secret Sharing

Let $\mathbb{F}$ be a finite field. Recall $t_c$ is the number of maliciously corrupted parties in each committee. A $(t_c + 1)$-out-of-$n$ secret sharing scheme takes as input a secret $z$ from $\mathbb{F}$ and outputs $n$ shares, one for each party, with the property that it is possible to efficiently recover $z$ from every subset of $t_c + 1$ shares, but every subset of at most $t_c$ shares reveals nothing about the secret $z$.

A secret sharing scheme consists of two algorithms: the first, Share, takes as input the secret $z$ and the parameters $n$ and $t_c$, and outputs $n$ shares: $(z^1, \ldots, z^n) = \text{Share}(z, n, t_c)$. We denote the vector of shares as $[z]_{t_c} = (z^1, \ldots, z^n)$. The second algorithm, Recons, takes as input a set of reconstructing parties $\Gamma \subseteq [n]$ and share $z^i$ and outputs a reconstruction value $\text{Recons}(\Gamma, z^i)$. We will utilize secret sharing schemes in which $\lambda_i \cdot z^i = \text{Recons}(\Gamma, z^i)$, for some constant $\lambda_i$ dependent on $i$ and $\Gamma$. If $|\Gamma| \geq t_c + 1$, then these reconstruction values can be simply summed to obtain $z = \sum_i \lambda_i \cdot z^i$. The secret sharing scheme we use is also *linear*, meaning that if the parties compute $[z_1]_{t_c} + [z_2]_{t_c}$, then invoke Recons to get reconstruction values $\lambda_i \cdot (z_1^i + z_2^i)$ for a big enough set $\Gamma$ of parties, summing these reconstruction values will yield $z_1 + z_2 = \sum_i \lambda_i(z_1^i + z_2^i)$. One instantiation of secret sharing uses a random degree $t_c$ polynomial $f(x)$, where the secret is stored at $f(0)$ and the share of each

$P_i$ is $f(i)$ (Shamir, 1979). Reconstruction uses polynomial interpolation, where the $\lambda_i$ are Lagrange coefficients.

Packed secret sharing is an extension of secret sharing, where a secret vector $\boldsymbol{z} = (z_1, \ldots, z_k) \in \mathbb{F}^k$ is *packed* into a single set of (individual) shares. We call $k$ the *packing parameter*. We still have that every subset of at most $t_c$ shares reveals nothing about $\boldsymbol{z}$, but we need at least $t_c + k$ shares to be able to recover $\boldsymbol{z}$. There are also similar Share and Recons algorithms, and we denote a sharing of some vector $\boldsymbol{z}$ as $[\boldsymbol{z}]_{t_c+k-1} = (\boldsymbol{z}^1, \ldots, \boldsymbol{z}^n)$. In addition, Recons takes as input an index $j \in [k]$ representing the index of the vector to reconstruct. We utilize packed secret sharing schemes in which $\lambda_i^j \cdot \boldsymbol{z}^i = \mathsf{Recons}(\Gamma, \boldsymbol{z}^i, j)$, for some constant $\lambda_i^j$. If $|\Gamma| \geq t_c + k$, then $z_j$ can be computed as $z_j = \sum_i \lambda_i^j \cdot \boldsymbol{z}^i$. The packed secret sharing scheme we use is also *linear* with respect to vector addition of the underlying secrets. One instantiation of packed secret sharing extends the polynomial idea from above—$f(x)$ is now degree $t_c + k - 1$ and each secret $z_j$ is stored at $f(-j)$; everything else stays the same (Franklin & Yung, 1992).

In the following, $t_c$ and $k$ will be fixed, so we will simply refer to packed secret sharings as $[\boldsymbol{z}]$.

## 3. Linear Secret Resharing Protocol

In this section, we present our constant-overhead linear secret resharing protocol, LRP.

**Naive $n^2$-Overhead Protocol.** We start with the naive $n^2$-overhead protocol, which follows from classical cryptographic literature (Ben-Or et al., 1988). Let it be the case that a secret sharing $[z]$ has been generated for parties in a given committee. To reshare this value to the parties of the next committee, each party $P_i$ in this committee distributes to them a sharing $[z^i]$ of their share. Since we know it is the case that $z = \sum_i \lambda_i \cdot z^i = \sum_i \mathsf{Recons}(\Gamma, z^i)$ and the secret sharing is linear, the parties in the next committee can simply compute their new sharing of $z$ to be $[z'] = \sum_i \lambda_i \cdot [z^i]$, and it is clear that $z' = z$ (the parties of this second committee must also know $\Gamma$, the subset of clients who did not drop out in the first committee). The problem with this protocol is of course that it has $n^2$ total communication overhead—each of $n$ parties has to distribute $n$ shares to the next committee.

**Our Constant-Overhead Protocol.** We can instead start by using packed secret sharing. Our resharing protocol is pictorially presented and summarized in Figure 3. It works by cleverly batching across many packed sharings. Our resharing protocol consists of four algorithms: it inherits the first algorithm Share from an underlying linear packed secret sharing scheme. Now, let it be the case that $k$ packed secret sharings $[\boldsymbol{z}_1], \ldots, [\boldsymbol{z}_k]$, for length-$k$ secret vectors $\boldsymbol{z}_1, \ldots, \boldsymbol{z}_k \in \mathbb{F}^k$, are distributed to the $n$ parties of itera-

tion $T$ (so there are $k^2$ total secrets). The next algorithm, called the *resharing algorithm*, Reshare, takes as input the $k$ packed shares of party $P_i$ of iteration $T$, which we denote as the vector $\boldsymbol{z}_{[1,k]}^i = (\boldsymbol{z}_1^i, \ldots, \boldsymbol{z}_k^i)$, and outputs $n$ fresh shares of this vector to the parties of iteration $T+1$: $[\boldsymbol{z}_{[1,k]}^i] = ((\boldsymbol{z}_{[1,k]}^i)^1, \ldots, (\boldsymbol{z}_{[1,k]}^i)^n) = \mathsf{Reshare}(\boldsymbol{z}_{[1,k]}^i)$. Next, the *recovery algorithm*, Recover, takes as input the set of dropout parties $\mathsf{Drop}_T$ of iteration $T$ and the reshared shares of non-dropout parties of iteration $T$ sent to party $P_j$ of iteration $T+1$, $(\boldsymbol{z}_{[1,k]}^{i_1})^j, \ldots, (\boldsymbol{z}_{[1,k]}^{i_{\tilde{n}}})^j$, for $i_1, \ldots, i_{\tilde{n}} \in [n] \backslash \mathsf{Drop}_T$, and outputs new shares of the original secret vectors $\boldsymbol{z}_1, \ldots, \boldsymbol{z}_k$ for party $P_j$: $(\hat{\boldsymbol{z}}_1^j, \ldots, \hat{\boldsymbol{z}}_k^j) = \mathsf{Recover}(\mathsf{Drop}_T, (\boldsymbol{z}_{[1,k]}^{i_1})^j, \ldots, (\boldsymbol{z}_{[1,k]}^{i_{\tilde{n}}})^j)$.[4] The last algorithm Recons is also inherited from the underlying linear packed secret sharing scheme.

We present protocol LRP below.

- $\mathsf{Reshare}(\boldsymbol{z}_{[1,k]}^i)$: Outputs $[\boldsymbol{z}_{[1,k]}^i] = \mathsf{Share}(\boldsymbol{z}_{[1,k]}^i)$.
- $\mathsf{Recover}(\mathsf{Drop}, (\boldsymbol{z}_{[1,k]}^{i_1})^j, \ldots, (\boldsymbol{z}_{[1,k]}^{i_{\tilde{n}}})^j)$: Computes $\hat{\boldsymbol{z}}_m^j = \sum_l \mathsf{Recons}([n] \backslash \mathsf{Drop}, (\boldsymbol{z}_{[1,k]}^{i_l})^j, m)$, for $m \in [k]$. Then outputs $(\hat{\boldsymbol{z}}_1^j, \ldots \hat{\boldsymbol{z}}_k^j)$

Now we observe how $\mathsf{Recover}(\cdot)$ outputs packed shares of the original secrets. Recall that $\mathsf{Recons}([n] \backslash \mathsf{Drop}, (\boldsymbol{z}_{[1,k]}^{i_l})^j, m) = \lambda_{i_l}^m \cdot (\boldsymbol{z}_{[1,k]}^{i_l})^j$, so we can re-write $\hat{\boldsymbol{z}}_m^j = \sum_l \lambda_{i_l}^m \cdot (\boldsymbol{z}_{[1,k]}^{i_l})^j$. Moreover, each $(\boldsymbol{z}_{[1,k]}^{i_l})^j$ is a share of vector $\boldsymbol{z}_{[1,k]}^{i_l} = (\boldsymbol{z}_1^{i_l}, \ldots, \boldsymbol{z}_k^{i_l})$ for a *linear* packed secret sharing scheme. Thus, we are computing new packed shares of the vectors $\sum_l \lambda_i^m \cdot (\boldsymbol{z}_1^{i_l}, \ldots, \boldsymbol{z}_k^{i_l})$. Each $\boldsymbol{z}_\ell^{i_l}$ was itself $P_{i_l}$'s share of vector $\boldsymbol{z}_\ell$. Thus the packed shares we are computing indeed share the vectors:

$$\sum_l \lambda_{i_l}^m \cdot (\boldsymbol{z}_1^{i_l}, \ldots, \boldsymbol{z}_k^{i_l})$$
$$= (\sum_l \mathsf{Recons}([n] \backslash \mathsf{Drop}, \boldsymbol{z}_1^{i_l}, m), \ldots,$$
$$\sum_l \mathsf{Recons}([n] \backslash \mathsf{Drop}, \boldsymbol{z}_k^{i_l}, m))$$
$$= (\boldsymbol{z}_{1,m}, \ldots, \boldsymbol{z}_{k,m}).$$

**Security.** It is clear that the output of $\mathsf{Reshare}()$ reveals nothing to the $t_c$ corrupted parties, since it just uses $\mathsf{Share}()$ of the underlying packed secret sharing scheme, that is secure against $t_c$ corrupted parties. Since the number of honest parties that do not dropout is at least $n - t_d - t_c > (1/2 + \mu)n$, it is clear that this protocol is resilient to the $t_d$ (honest) dropout parties, if $k \leq 2\mu n$. This is because $t_c + k \leq (1/2 + \mu)n < n - t_d - t_c$, so the shares of the parties that do not dropout can still be used to reconstruct

---

[4]Note: the output shares are for length-$k$ secret vectors $(z_{1,m}, \ldots, z_{k,m})$ for each $m \in [k]$, instead of $(z_{\ell,1}, \ldots, z_{\ell,k})$, for each $\ell \in [k]$.

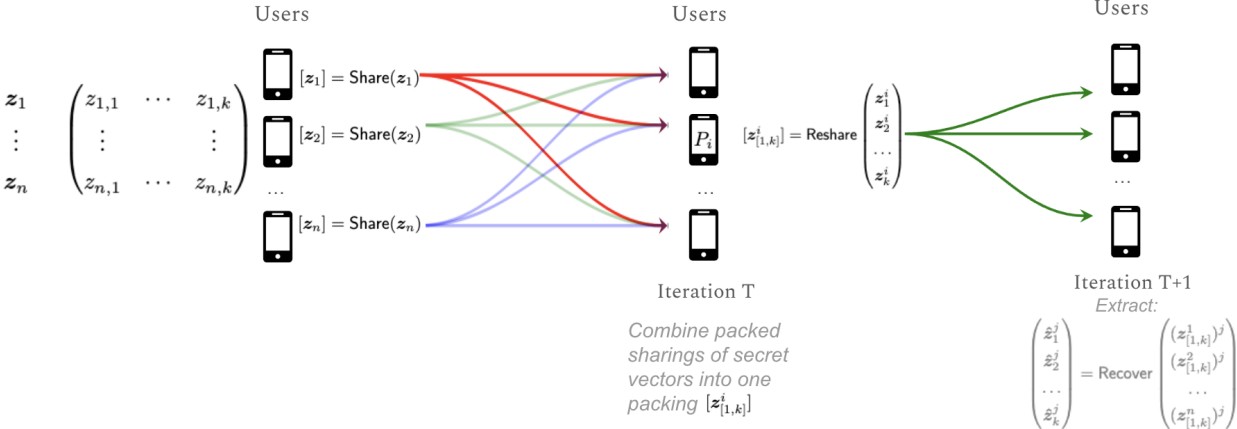

*Figure 3.* Constant-Overhead Linear Secret Resharing Protocol, LRP. At a high level, parties $P_i$ in iteration $T$ each receive packed secret sharings $[\mathbf{z}_i]$. Then, the parties of iteration $T$ reshare these packed shares by distributing packed sharings *of these packed shares* to the parties of iteration $T + 1$, who finally recover packed shares of the original secret vectors $\mathbf{z}_1, \ldots, \mathbf{z}_n$.

in the secret space during Recover. The fact that malicious parties can only cause reconstructed values to be perturbed by an independent value $\chi$ follows from standard facts about secret sharing (Genkin et al., 2014). We formally prove the security and dropout-resiliency of LRP in Section C.

**Communication Complexity.** Let $k = 2\mu n$. The communication cost of both Share and Recons is $n$ field elements for $k$ secrets, which is $1/2\mu$ per secret. The total communication complexity of Reshare is $n^2$ field elements—each party sends a share to every party in the next iteration. This is for $k^2 = 4\mu^2 n^2$ secrets, which is $1/4\mu^2$ per secret. Thus, the communication is at most $1/4\mu^2$ per secret ($\mu < 1/2$).

## 4. Distributed Matrix Mechanism

We now present our Distributed Matrix Mechanism. See Protocol 1 for a detailed description of $\Pi_{\mathsf{DMM}}$. We will assume that each committee has the same number $n$ of clients. We write the protocol in terms of batches of gradients of size $k^2$, where $k$ is the packing parameter; if the model dimension $d > k^2$, then the parties repeat $\Pi_{\mathsf{DMM}}$ over batches.[5]

Each iteration of this protocol is completed in two communication rounds.[6] In the $T$-th iteration, we will assume that the $n$ clients selected have received, for $\tau \in [T-1]$, (i) $[\mathbf{X}^{i_l}_{\tau,[1,k]}]$, which are the (aggregated) gradient shares from the first $T-1$ iterations, reshared by non-dropout party $i_l$ in the previous iteration; and (ii) $[\mathbf{Z}^{i_l}_{\tau,[1,k]}]$, which are the (aggregated) noise shares sampled in the first $T-1$ iterations,

reshared by party $i_l$ in the previous iteration. The clients first recover shares of the same:

$$(\hat{\mathbf{Z}}^j_{\tau,1}, \ldots, \hat{\mathbf{Z}}^j_{\tau,k}) =$$
$$\mathsf{Recover}(\mathsf{Drop}_{T-1}, (\mathbf{Z}^{i_1}_{\tau,[1,k]})^j, \ldots, (\mathbf{Z}^{i_{\bar{n}}}_{\tau,[1,k]})^j)$$

$$(\hat{\mathbf{X}}^j_{\tau,1}, \ldots \hat{\mathbf{X}}^j_{\tau,k}) =$$
$$\mathsf{Recover}(\mathsf{Drop}_{T-1}, (\mathbf{X}^{i_1}_{\tau,[1,k]})^j, \ldots, (\mathbf{X}^{i_{\bar{n}}}_{\tau,[1,k]})^j),$$

based on iteration $T-1$ dropout clients $\mathsf{Drop}_{T-1}$ received from the server.

Next, as in the distributed setting, the clients will compute their local gradients $\mathbf{g}_{T,i}$ (clipped, scaled, flattened, and rounded as in (Kairouz et al., 2021a)) and sample $\mathbf{z}_{T,i}$ from a noise distribution $\mathcal{D}$. Then, each client will compute some secret shares $[\mathbf{z}_{T,i}], [\mathbf{g}_{T,i}]$ of their local gradients and noise and distribute them to the other clients of this iteration. Once receiving these shares, the parties (locally) aggregate them: $[\mathbf{Z}_T] = (\sum_{\eta=1}^{n} [\mathbf{z}_{T,\eta}])$ and $[\mathbf{X}_T] = (\sum_{\eta=1}^{n} [\mathbf{g}_{T,\eta}])$.

The parties then take linear combinations, according to $\mathbf{A}$ and $\mathbf{B}$, of the packed shares of gradients and noise of all previous iterations, including this one, to obtain shares of the next output of the matrix mechanism, $[\widehat{\mathbf{A}\mathbf{X}}_T]$. The parties then reconstruct these noisy gradients to the server (which are then unflattened and rescaled by the server (Kairouz et al., 2021a)).

Finally, the clients will reshare their shares $\hat{\mathbf{Z}}^j_{\tau,[1,k]}$ and $\hat{\mathbf{X}}^j_{\tau,[1,k]}$ of the aggregated noise and gradients from the first $T$ iterations. The clients reshare the shares according to protocol LRP in Section 3.

**An Optimization.** Typically, the matrix $\mathbf{A}$ is just the prefix sum matrix. In this case, the server anyway sees $\widehat{\mathbf{A}\mathbf{X}}_T - \widehat{\mathbf{A}\mathbf{X}}_{T-1} = \mathbf{X}_T + (\mathbf{B}_{[T:,]} - \mathbf{B}_{[T-1:,]})\mathbf{Z}$, so we can just

---

[5]We also assume that communication between clients in $\Pi_{\mathsf{DMM}}$ is done via authenticated and encrypted channels, routed through the server and using a Public-Key Infrastructure (PKI), as in previous works, e.g., (Bonawitz et al., 2017), etc.

[6]Note that if there are no dropouts, each iteration can complete in one communication round.

---

**Protocol 1** Differentially-Private Federated Learning Protocol $\Pi_{\mathsf{DMM}}$

---

**Subprotocols**: LRP = (Share, Reshare, Recons, Recover) is a secret resharing protocol (See Section 3).
**Parameters:** Packing parameter $k \in \mathbb{N}$; number of iterations $T^*$; finite field $\mathbb{F}$ of bit-width $m$; matrix $\boldsymbol{A} \in \mathbb{R}^{T^* \times T^*}$ and $\boldsymbol{B}, \boldsymbol{C}$ such that $\boldsymbol{A} = \boldsymbol{BC}$; noise distribution $\mathcal{D}$.
**Inputs:** Current iteration $T$; gradients $\boldsymbol{g}_{T,j,1}, \ldots, \boldsymbol{g}_{T,j,k} \in \mathbb{F}^k$ ; list of dropped clients from iteration $T-1$, $\mathsf{Drop}_{T-1}$ (If $|\mathsf{Drop}_{T-1}| < t_c + k$, then abort); reshared gradients and noise received from $\mathcal{C}_{T-1}$ for the first $T-1$ iterations $\{[\boldsymbol{X}_{\tau,[1,k]}^{i_l}], [\boldsymbol{Z}_{\tau,[1,k]}^{i_l}]\}_{\tau \in [T-1], i_l \in [n] \setminus \mathsf{Drop}_{T-1}}$.

Round 1:

    **Parties $P_j$:**
- Sample noise vectors $\boldsymbol{z}_{T,j,1}, \ldots, \boldsymbol{z}_{T,j,k} \in \mathbb{F}^k$ from $\mathcal{D}$.
- For each $\ell \in [k]$, distribute packed secret sharings $[\boldsymbol{z}_{T,j,\ell}] = \mathsf{Share}(\boldsymbol{z}_{T,j,\ell})$ and $[\boldsymbol{g}_{T,j,\ell}] = \mathsf{Share}(\boldsymbol{g}_{T,j,\ell})$ to the set $\mathcal{C}_T$ of clients of this training iteration (via authenticated and encrypted channels through the server).

    **Server**:
- Receive from each Party $P_j$ in $\mathcal{C}_T$ and register dropped clients in list $\mathsf{Drop}_T$.
- Forward (encrypted) shares $[\boldsymbol{z}_{T,j,\ell}]$ and $[\boldsymbol{g}_{T,j,\ell}]$ to the other clients of $\mathcal{C}_T$, along with $\mathsf{Drop}_T$.

Round 2:

    **Parties $P_j$:**
- Receive from server the list of dropped clients from iteration $T$, $\mathsf{Drop}_T$.
- For each $\ell \in [k]$, aggregate $[\hat{\boldsymbol{Z}}_{T,\ell}] = (\sum_{\eta \in [n] \setminus \mathsf{Drop}_T}[\boldsymbol{z}_{T,\eta,\ell}])$ and $[\hat{\boldsymbol{X}}_{T,\ell}] = (\sum_{\eta \in [n] \setminus \mathsf{Drop}_T}[\boldsymbol{g}_{T,\eta,\ell}])$; then reshare $[\boldsymbol{Z}_{T,[1,k]}^j] = \mathsf{Reshare}(\hat{\boldsymbol{Z}}_{T,[1,k]}^j)$ and $[\boldsymbol{X}_{T,[1,k]}^j] = \mathsf{Reshare}(\hat{\boldsymbol{X}}_{T,[1,k]}^j)$ to the set of clients in $\mathcal{C}_{T+1}$ (via authenticated and encrypted channels through the sever).
- **If $T = 1$:**
    - For each $\ell \in [k]$, compute $[\boldsymbol{Y}_{1,\ell}] = \boldsymbol{A}_{[1,1]} \cdot [\hat{\boldsymbol{X}}_{1,\ell}] + \boldsymbol{B}_{[1,1]} \cdot [\hat{\boldsymbol{Z}}_{1,\ell}]$, then send shares $\boldsymbol{Y}_{1,\ell}^j$ to the server.
- **If $T > 1$:**
    - For $\tau \in [T-1]$, recover $(\hat{\boldsymbol{Z}}_{\tau,1}^j, \ldots, \hat{\boldsymbol{Z}}_{\tau,k}^j) = \mathsf{Recover}(\mathsf{Drop}_{T-1}, (\boldsymbol{Z}_{\tau,[1,k]}^{i_1})^j, \ldots, (\boldsymbol{Z}_{\tau,[1,k]}^{i_{\tilde{n}}})^j)$ and $(\hat{\boldsymbol{X}}_{\tau,1}^j, \ldots \hat{\boldsymbol{X}}_{\tau,k}^j) = \mathsf{Recover}(\mathsf{Drop}_{T-1}, (\boldsymbol{X}_{\tau,[1,k]}^{i_1})^j, \ldots, (\boldsymbol{X}_{\tau,[1,k]}^{i_{\tilde{n}}})^j)$ to obtain shares $[\hat{\boldsymbol{Z}}_{\tau,\ell}]$ and $[\hat{\boldsymbol{X}}_{\tau,\ell}]$ for $\ell \in [k]$.
    - Then again reshare as $[\boldsymbol{Z}_{\tau,[1,k]}^j] = \mathsf{Reshare}(\hat{\boldsymbol{Z}}_{\tau,[1,k]}^j)$ and $[\boldsymbol{X}_{\tau,[1,k]}^j] = \mathsf{Reshare}(\hat{\boldsymbol{X}}_{\tau,[1,k]}^j)$ to set $\mathcal{C}_{T+1}$ (via authenticated and encrypted channels through the sever).
    - For each $\ell \in [k]$, compute $[\boldsymbol{Y}_{T,\ell}] = \sum_{\tau=1}^{T} \boldsymbol{A}_{[T,\tau]} \cdot [\hat{\boldsymbol{X}}_{\tau,\ell}] + \boldsymbol{B}_{[T,\tau]} \cdot [\hat{\boldsymbol{Z}}_{\tau,\ell}]$, then send shares $\boldsymbol{Y}_{T,\ell}^j$ to the server.

    **Server:**
- Receive from each Party $P_j$ in $\mathcal{C}_T$ and register dropped clients in list $\mathsf{Drop}_T$.
- For each $\ell, m \in [k]$, compute and output $Y_{T,\ell,m} = \sum_{j \in [n] \setminus \mathsf{Drop}_T} \mathsf{Recons}([n] \setminus \mathsf{Drop}_t, \boldsymbol{Y}_{T,\ell}^j, m)$.
- Forward (encrypted) reshares to the clients of $\mathcal{C}_{T+1}$ and send dropped clients list $\mathsf{Drop}_T$ to clients of $\mathcal{C}_{T+1}$

---

reconstruct this to it, and do not need to reshare the gradients $\boldsymbol{X}$ across iterations, saving a factor of two. This is how we compute the communication complexity of our protocol.

**Matrix Factorizations and Communication Complexity.** We use two different matrix factorizations $\boldsymbol{A} = \boldsymbol{BC}$ for our experiments. The first is the optimal with respect to the loss function $\mathcal{L}(\boldsymbol{B}, \boldsymbol{C}) = \mathsf{sens}_\Phi(\boldsymbol{C})\|\boldsymbol{B}\|_F^2$ for the $b$-min-sep-participation schema $\Phi$ (Choquette-Choo et al., 2023a). In this case, in iteration $T$, the total communication complexity is $(d \cdot T)/(4\mu^2)$, using $k = 2\mu n$ as above. The second factorization is the Honaker Online mechanism (Kairouz

et al., 2021b; Honaker, 2015), where $\boldsymbol{C}$ is essentially the binary tree matrix.[7] This mechanism has the benefit that it allows for implementations with only $(d \log T)/(4\mu^2)$ total communication complexity; in the $T$-th iteration, the released model can be computed by a sum of at most $d \cdot \log(T)$ sharings.

**Security.** We formally prove the security of $\Pi_{\mathsf{DMM}}$ in Section C based on the security of LRP, i.e., nothing but the noisy gradients are revealed to an adversary corrupting at

---

[7]$\Pi_{\mathsf{DMM}}$ can easily use any factorization, including BLTs (McMahan et al., 2024), which are not open sourced.

most $t_c$ parties in each iteration and the server. We also prove dropout tolerance and distributed DP even in the presence of corrupted parties that perturb the noisy gradients released to the server by independent values $\chi$.

**Privacy.** We now state the privacy of our protocol when the noise distribution $\mathcal{D}$ is the Discrete Gaussian distribution with mean 0 and variance $\sigma^2/\gamma^2$, $\mathcal{N}_{\mathbb{Z}}(0, \sigma^2/\gamma^2)$.[8] We note that another option for noise is the Skellam Distribution (Agarwal et al., 2021) that yields roughly the same privacy-utility tradeoff; thus we stick to the more standard Discrete Gaussian. Moreover, our preliminary experiments showed that the Skellam Distribution did not seem to perform even close to as well empirically as the Discrete Gaussian. Another tempting choice to obtain DP is to use the technique of the Poisson Binomial Mechanism (Chen et al., 2022), however, that work departs from the additive noise paradigm and thus does not seem applicable for us.

First we explain some parameters: $c$ is the norm to which gradients are clipped, $\gamma > 0$ is used to determine the granularity for the discretization of gradients, $\beta$ determines the bias of the randomized rounding for discretization, and $\sigma$ is the noise scale of the Discrete Gaussians. Details on these steps (from (Kairouz et al., 2021a)) are provided in Section A. The following theorem is proved in Section B.

**Theorem 4.1.** *Consider a query matrix $\boldsymbol{A} \in \mathbb{R}^{T^* \times T^*}$ along with a fixed factorization $\boldsymbol{A} = \boldsymbol{B}\boldsymbol{C}$ with $\Delta = \mathrm{sens}_\Phi^1(\boldsymbol{C})$. Let $\tau := 10 \cdot \sum_{k=1}^{n-1} e^{-2\pi^2 \frac{\sigma^2}{\gamma^2} \cdot \frac{k}{k+1}}$ and*

$$\hat{c}^2 := \min \left\{ \begin{array}{l} c^2 + \dfrac{\gamma^2}{4} d + \sqrt{2\log(1/\beta)} \cdot \gamma \cdot (c + \dfrac{\gamma}{2}\sqrt{d}), \\ (c + \gamma\sqrt{d})^2 \end{array} \right\},$$

*Then $\Pi_{\mathsf{DMM}}$ satisfies $\frac{1}{2}\varepsilon^2$-concentrated DP for*

$$\varepsilon := \min \left\{ \sqrt{\dfrac{\Delta^2 \hat{c}^2}{n\sigma^2} + 2\tau d}, \dfrac{\Delta\hat{c}}{\sqrt{n}\sigma} + \tau\sqrt{d} \right\}.$$

**Accuracy.** We now formally prove the accuracy of our Distributed Matrix Mechanism (DMM). First, we explain an additional parameter: $m$ is the bit-width of the finite field $\mathbb{F}$ used in $\Pi_{\mathsf{DMM}}$. We prove the following in Section B.

**Theorem 4.2.** *Let $n, m, d, T^* \in \mathbb{N}$, and $c, \varepsilon > 0$ satisfy:*

$$m \geq \tilde{O}\left( \max_{T \in [T^*]} ||\boldsymbol{A}_{[T,:]}||_2 \sqrt{nT} + \max_{T \in [T^*]} ||\boldsymbol{B}_{[T,:]}||_2 \frac{\sqrt{d}\Delta}{\varepsilon} \right).$$

*Let $\Pi_{\mathsf{DMM}}$ be instantiated with parameters*
$$\gamma = \tilde{O}\left( \frac{\max_{T \in [T^*]} ||\boldsymbol{A}_{[T,:]}||_2 c\sqrt{nT}}{m\sqrt{d}} + \frac{\max_{T \in [T^*]} ||\boldsymbol{B}_{[T,:]}||_2 c\Delta}{\varepsilon m} \right),$$
$$\beta \leq \Theta\left(\frac{1}{n}\right), \text{ and } \sigma = \tilde{\Theta}\left( \frac{c\Delta}{\varepsilon\sqrt{n}} + \sqrt{\frac{d}{n}} \cdot \frac{\gamma\Delta}{\varepsilon} \right).$$

*Then $\Pi_{\mathsf{DMM}}$ attains $\frac{1}{2}\varepsilon^2$-concentrated DP and accuracy:*

$$\sum_{T=1}^{T^*} \mathbb{E}\left[ \left|\left| \Pi_{\mathsf{DMM}}(X) - \boldsymbol{A}_{[T,:]} \sum_{i=1}^{n} \boldsymbol{X}_i \right|\right|_2^2 \right]$$
$$\leq O\left( ||\boldsymbol{B}||_F^2 \frac{c^2\Delta^2 d}{\varepsilon^2} \right).$$

## 5. Experiments

Here we empirically evaluate DMM for FL on the Stack Overflow Next Word Prediction (SO-NWP) (Authors, 2019) and FEMNIST (Caldas et al., 2018) public benchmarks. We compare to the prior state-of-the art for privacy-utility tradeoff with distributed DP, the Distributed Discrete Gaussian Mechanism (DDG) (Kairouz et al., 2021a) which also uses privacy amplification via sampling (DMM does not), and central DP, BandMF (Choquette-Choo et al., 2023a). Our full experimental setup is described in Section D, and closely follows prior work, including model hyperparameters (Kairouz et al., 2021a; Choquette-Choo et al., 2023a). All experiments are run on a machine with an AMD EPYC 7R32 processor and an A10G GPU. See Section D for further evaluation of our results.

**Privacy Parameters and Selected Hyperparameters.** For both matrix factorizations, we measure $\mathrm{sens}_\Phi^1(\boldsymbol{C})$ with respect to the $b$-min-sep-participation schema using Choquette-Choo et al. (2023a, Theorems 2 and 3). For SO-NWP, we use $T^* = 2052$ and $b = 342$ (as in (Choquette-Choo et al., 2023a)) and we use $T^* = 2^{11} = 2048$ and $b = 512$ for the Honaker factorization, since $T^*$ needs to be a power of two. For FEMNIST, we use $T^* = 1445$ (similar to (Kairouz et al., 2021a)) and $b = 85$ for the optimal factorization and $T^* = 2^{10} = 1024$ and $b = 64$ for the Honaker factorization—the reason for smaller bands is that there is less data in the FEMNIST dataset, which means clients have to participate more often.

**Results.** For performance evaluation of DMM, we use $n = 40$ clients per iteration. Figures 2 and 4 show that across $\varepsilon$ privacy levels, our DMM significantly outperforms DDG in terms of classification accuracy. This is precisely because DMM uses correlated noise across iterations, whereas DDG uses fresh noise in each iteration. Our DMM also gets accuracy close to that of the state-of-the-art central DP solutions for SO-NWP. This gap comes from the error introduced in discretizing values and modular clipping that both arise from DMM representing numbers as finite field elements, as well as some error arising from summing Discrete Gaussians (see Section B). These errors are also present in DDG (Kairouz et al., 2021a). We also see that using the Honaker factorization only slightly degrades the accuracy compared to the mechanism based on the optimal $b$-min-sep-participation matrix factorization. Therefore, the

---

[8]Note that, as with prior work in the distributed DP setting, e.g., (Kairouz et al., 2021a), we must use discrete noise. This is because cryptographic techniques work over finite, discrete algebraic structures (like finite fields/rings/groups).

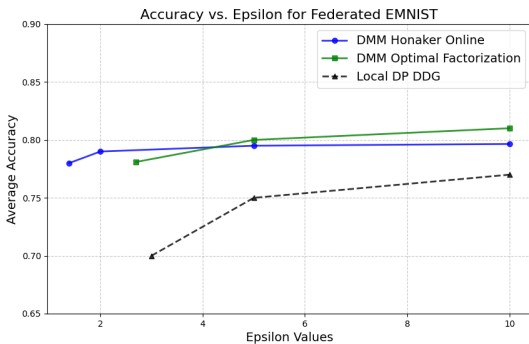

| | LRP Comp. | SecAgg Comp. | LRP Comm. | Naive SR Comm. | SecAgg Comm. |
|---|---|---|---|---|---|
| Opt. | 3.34 s | 58.5 ms | 828 MB | 379 GB | 4.07 MB |
| Hon. | 94.2 ms | 58.5 ms | 5.73 MB | 2.61 GB | 4.07 MB |

*Table 2.* Client computation and communication of our LRP resharing protocol, naive secret resharing, and SecAgg per iteration on Federated EMNIST for committee size $n = 64$. We give results for both the optimal and more efficient Honaker online matrix mechanisms. (ms := milliseconds; s := seconds).

*Figure 4.* Test accuracies on FEMNIST across different privacy levels $\varepsilon$ for the distributed DP DDG mechanism and our distributed DP DMM instantiated with the optimal matrix factorization and the Honaker online matrix factorization. DMM performs $\approx 4$ percentage points better than the prior distributed DP approach. We use $\delta = 1/N$ for $(\varepsilon, \delta)$-DP, where $N$ is the total number of clients across training.

tree mechanism might be best in practice due to much better efficiency, seen below.

**Efficiency.** Tables 1 and 2 show client computation and communication costs of DMM for SO-NWP and FEMNIST, respectively, using both the optimal matrix factorization and the Honaker matrix factorization. We show the costs of SecAgg (the bottleneck of DDG) using the Flamingo construction (Ma et al., 2023) (recall: any SecAgg protocol can be used; Flamingo is the state-of-the-art) and the costs of the resharing protocol LRP in $\Pi_{\mathsf{DMM}}$. We also give the communication costs of the naive secret resharing protocol of Section 3. For $\Pi_{\mathsf{DMM}}$, we assume $\mu = 1/6$; i.e., the number of corrupted and dropout parties per iteration satisfies $t_c + t_d < n/3$. We use 32 bits to represent field values. For computational experiments, we use $n = 64$, as the Flamingo code requires powers of two. For the optimal matrix factorization results, we report the worst-case complexity per iteration, which is the penultimate iteration, since clients reshare the noise from all previous iterations.

We see that the naive $n^2$ overhead secret resharing protocol has infeasible communication of up to 2.13 TB per client, which is substantially more than the communication of LRP, with communication as low as 5.73 MB per client. We also see that the optimal matrix factorization substantially increases both the computation and communication compared to Honaker online factorization. This suggests that the small increase in accuracy from using the optimal matrix factorization may not be worth it in terms of the added efficiency costs. Compared to SecAgg used by DDG, we see a modest increase in computation with the Honaker online factorization from LRP in $\Pi_{\mathsf{DMM}}$; less than 10 seconds (and sometimes less than 100 ms) per iteration is very reasonable.

In terms of communication, we see an increase of $< 10$ MB with the Honaker online factorization from LRP in $\Pi_{\mathsf{DMM}}$ compared to that of SecAgg in DDG. We believe that this added overhead is worth it given the increased accuracy.

## 6. Conclusion

We present in this paper the Distributed Matrix Mechanism (DMM) for FL, which achieves both distributed DP and privacy-utility trade-off of the matrix mechanism for central DP in FL. Along the way, we introduce a constant-overhead linear secret resharing protocol LRP. We validate experimentally the utility and efficiency of DMM. Future work includes designing better low-memory matrix factorizations to get efficiency with better accuracy, as well as adding malicious security to the *encryption* approach of (Ball et al., 2024).

## Acknowledgements

We thank Keith Rush for providing valuable assistance with the code used to factorize matrices optimally.

## Disclaimer

## Impact Statement

Our work provides privacy for FL using formal $(\varepsilon, \delta)$-DP guarantees. One should ensure when using DP that the used $(\varepsilon, \delta)$ privacy levels are adequate for protecting sensitive data in their setting.

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

**Output**: $\gamma \cdot U^\intercal \widehat{AX}_T' \in \mathbb{R}^d$.

---

# Supplementary Material

## A. Discretization Details of (Kairouz et al., 2021a)

We use the randomized rounding strategy from (Kairouz et al., 2021a) for discretization in $\Pi_{\mathsf{DMM}}$. At a high-level, each client first clips and scales their input gradient. Then, the clients flatten their gradient vectors using some unitary matrix $U$ (intuitively, this minimizes the risk of modulo overlap in vector elements that are particularly large). Finally, the clients use a randomized process to round their gradient vectors in $\mathbb{R}^d$ to $\mathbb{Z}^d$. On the sever side, after receiving the aggregated, noise outputs $\hat{AX}_T$ in each round, the server unflattens the vector by applying $U^T$ and then descales. Protocols 2 and 3 give more detai, but we refer the readers to (Kairouz et al., 2021a) for full details on possible flattening matrices $U$ and the randomized rounding procedure used.

To help with the analysis, (Kairouz et al., 2021a) uses the following definitions to represent the conditional randomized rounding. We present them verabtim.

**Definition A.1** (Randomized Rounding). Let $\gamma > 0$ and $d \in \mathbb{N}$. Define $R_\gamma : \mathbb{R}^d \to \gamma\mathbb{Z}^d$ (where $\gamma\mathbb{Z}^d := \{(\gamma z_1, \gamma z_2, \ldots, \gamma z_d) : z_1, \ldots, z_d \in \mathbb{Z}\} \subseteq \mathbb{R}^d$) as follows. For $x \in [0, \gamma]^d$, $R_\gamma(x)$ is a product distribution on $\{0, \gamma\}^d$ with mean $x$; that is, independently for each $i \in [d]$, we have $\Pr[R_\gamma(x)_i = 0] = 1 - x_i\gamma$ and $\Pr[R_\gamma(x)_i = \gamma] = x_i/\gamma$. In general, for $x \in \mathbb{R}^d$, we have $R_\gamma(x) = \gamma\lfloor x/\gamma \rfloor + R_\gamma(x - \gamma\lfloor x/\gamma \rfloor)$; here $\gamma\lfloor x/\gamma \rfloor \in \gamma\mathbb{Z}^d$ is the point $x$ rounded down coordinate-wise to the grid.

**Definition A.2** (Conditional Randomized Rounding). Let $\gamma > 0$ and $d \in \mathbb{N}$ and $G \subseteq \mathbb{R}^d$. Define $R_\gamma^G : \mathbb{R}^d \to \gamma\mathbb{Z}^d \cap G$ to be $R_\gamma$ conditioned on the hte output being in $G$. That is, $\Pr[R_\gamma^G(x) = y] = \Pr[R\gamma(x) = y]/\Pr[R_\gamma(x) \in G]$ for all

$y \in \gamma \mathbb{Z}^d \cap G$, where $R_\gamma$ is as in Definition A.1.

## B. Proofs for Section 4

### B.1. Proof of Theorem 4.1

First we recall the notion of Rényi Divergences and Concentrated Differential Privacy (Bun & Steinke, 2016; Dwork & Rothblum, 2016), as well as some other standard DP notions. We also define the Discrete Gaussian and provide its DP guarantees. See (Kairouz et al., 2021a) for more details. Then we prove Thoerem 4.1

**Definition B.1** (Rényi Divergences). Let $P$ and $Q$ be probability distributions on some common domain $\Omega$. Assume that $P$ is absolutely continuous with respect to $Q$ so that the Radon-Nikodym derivative $P(x)/Q(x)$ is well-defined for $x \in \Omega$.

For $\alpha \in (1, \infty)$, we define the Rényi Divergence of order $\alpha$ of $P$ with respect to $Q$ as:

$$D_\alpha(P||Q) := \frac{1}{\alpha - 1} \log \mathbb{E}_{X \leftarrow P} \left[ \left( \frac{P(X)}{Q(x)} \right)^{\alpha - 1} \right]$$

We also define

$$D_*(P||Q) := \sup_{\alpha \in (1,\infty)} \frac{1}{\alpha} D_\alpha(P||Q)$$

**Definition B.2** (Concentrated Differential Privacy (Bun & Steinke, 2016; Dwork & Rothblum, 2016)). A randomized algorithm $M : \mathcal{X}^* \to \mathcal{Y}$ satisfies $\frac{1}{2}\varepsilon$-concentrated differential privacy iff, for all $x, x' \in \mathcal{X}$ differing by the addition or removal of a single user's records, we have $D_*(M(x)||M(x')) \leq \frac{1}{2}\varepsilon^2$.

**Definition B.3** (Rényi Differential Privacy (Mironov, 2017)). A randomized algorithm $M : \mathcal{X}^* \to \mathcal{Y}$ satisfies $(\alpha, \varepsilon)$-Rényi differential privacy iff, for all $x, x' \in \mathcal{X}$ differing by the addition or removal of a single user's records, we have $D_\alpha(M(x)||M(x')) \leq \frac{1}{2}\varepsilon^2$.

**Definition B.4** (Differential Privacy (Dwork et al., 2006)). A randomized algorithm $M : \mathcal{X}^* \to \mathcal{Y}$ satisfies $(\varepsilon, \delta)$-differential privacy iff, for all $x, x' \in \mathcal{X}$ differing by the addition or removal of a single user's records, we have

$$\Pr[M(x) \in E] \leq e^\varepsilon \Pr[M(x') \in E] + \delta,$$

for all events $E \subset Y$. We refer to $(\varepsilon, 0)$-DP as pure DP and $(\varepsilon, \delta)$-DP for $\delta > 0$ as approximate DP.

We remark that $\frac{1}{2}\varepsilon^2$-concentrated DP is equivalent to satisfying $(\alpha, \frac{1}{2}\varepsilon^2\alpha)$-Rényi DP simultaneously for all $\alpha \in (1, \infty)$. We also have the following conversion lemma from concentrated to approximate DP (Balle et al., 2020; Canonne et al., 2020; Asoodeh et al., 2020).

**Lemma B.5.** *If $M$ satisfies $(\varepsilon, 0)$-DP, then it satisfies $\frac{1}{2}\varepsilon^2$-concentrated DP. If $M$ satisfies $\frac{1}{2}\varepsilon^2$-DP then, for any $\delta > 0$, $M$ satisfies $(\varepsilon_{aDP}(\delta), \delta)$-DP, where*

$$\varepsilon_{aDP}(\delta) = \inf_{\alpha > 1} \frac{1}{2}\varepsilon^2\alpha + \frac{\log(1/\alpha\delta)}{\alpha - 1} + \log(1 - 1/\alpha) \leq \varepsilon \cdot (\sqrt{2\log(1/\delta)} + \varepsilon/2).$$

**Discrete Gaussian**   Here we define the Discrete Gaussiasn (Canonne et al., 2020) and give its DP guarantees.

**Definition B.6** (Discrete Gaussian). The discrete Gaussian with scale parameter $\sigma > 0$ and location parameter $\mu \in \mathbb{Z}$ is a probability distribution supported on the integers $\mathbb{Z}$ denoted by $\mathcal{N}_{\mathbb{Z}}(\mu, \sigma^2)$ and defined by

$$\forall x \in \mathbb{Z} \quad \Pr_{X \leftarrow \mathcal{N}_{\mathbb{Z}}(\mu, \sigma^2)}(X = x) = \frac{\exp\left( \frac{-(x-\mu)^2}{2\sigma^2} \right)}{\sum_{y \in \mathbb{Z}} \exp\left( \frac{-(y-\mu)^2}{2\sigma^2} \right)}.$$

**Proposition B.7** ((Kairouz et al., 2021a), Proposition 14). *Let $\sigma \geq \frac{1}{2}$. Let $X_{I,j} \leftarrow \mathcal{N}_{\mathbb{Z}}(0, \sigma^2)$ independently for each $i$ and*

$j$. Let $X_i = (X_{i,1}, \ldots, X_{i,d}) \in \mathbb{Z}^d$. Let $Z_n = \sum_{i=1}^n X_i \in \mathbb{Z}^d$. Then, for all $\Delta \in \mathbb{Z}^d$ and all $\alpha \in (1, \infty)$,

$$D_\alpha(Z_n \| Z_n + \Delta) \leq \min\{ \frac{\alpha \|\Delta\|_2^2}{2n\sigma^2} + \tau d,$$
$$\frac{\alpha}{2} \cdot \left( \frac{\|\Delta\|_2^2}{n\sigma^2} + 2\frac{\|\Delta\|_1}{\sqrt{n}\sigma} \cdot \tau + \tau^2 d \right),$$
$$\frac{\alpha}{2} \cdot \left( \frac{\|\Delta\|_2}{\sqrt{n}\sigma} + \tau\sqrt{d} \right)^2 \}$$

where $\tau := 10 \cdot \sum_{k=1}^n e^{-2\pi^2 \sigma^2 \frac{k}{k+1}}$. An algorithm $M$ that adds $Z_n$ to a query with $\ell_p$ sensitivity $\Delta_p$ satisfies $\frac{1}{2}\varepsilon^2$-concentrated DP for

$$\varepsilon = \min\{ \sqrt{\frac{\|\Delta\|_2^2}{n\sigma^2} + 2\tau d},$$
$$\sqrt{\frac{\Delta_2^2}{n\sigma^2} + 2\frac{\Delta_1}{\sqrt{n}\sigma} \cdot \tau + \tau^2 d},$$
$$\frac{\Delta_2}{\sqrt{n}\sigma} + \tau\sqrt{d} \}$$

### Proof of Theorem 4.1

*Proof.* First, it is sufficient to show that the computation $CG + Z$ satisfies $\frac{1}{2}\varepsilon^2$-concentrated DP, due to the post processing property of DP. Now consider two datasets $G$ and $H$ differing in one data record according to participation schema $\Phi$.[9] By assumption in the theorem statement, we have

$$\text{sens}_\Phi^1(C) = \Delta, \quad \text{and thus} \quad \text{sens}_\Phi(C) = c' \cdot \Delta,$$

where $c'$ is the bound on the $\ell_2$ norm of individual gradient vectors that are aggregated. Since we use the randomized rounding techniques from Section A, gradients that are clipped to $\ell_2$ norm $c$ can actually end up having $\ell_2$ norm $c' = \hat{c}$ after rounding, where $\hat{c}$ is as in the theorem statement. With the bound on the total sensitivity above, we know from Kairouz et al. (2021a, Proposition 14) (reproduced above) that the computation is $\frac{1}{2}\varepsilon^2$-concentrated DP, with the $\varepsilon$ from the theorem statement. $\square$

### B.2. Proof of Theorem 4.2

We first prove the following exact result for the error:

**Theorem B.8.** *Let* $\beta \in [0, 1)$, $\sigma^2 \geq \gamma/2 > 0$, *and* $c > 0$. *Let* $n, d \in \mathbb{N}$ *and* $\rho \geq 1$. *Let* $g_{T,i} \in \mathbb{R}^d$ *with* $\|g_{T,i}\|_2 \leq c$ *for each* $T \in [T^*], i \in [n]$. *Let* $U \in \mathbb{R}^{d \times d}$ *be a random unitary matrix such that*

$$\forall \boldsymbol{x} \in \mathbb{R}^d \quad \forall i \in [d] \quad \forall t \in \mathbb{R} \quad \mathbb{E}[\exp(t(Ux)_i)] \leq \exp(t^2 \rho \|x\|_2^2 / 2d).$$

*Let*

$$\Delta = \text{sens}_\Phi^1(C)$$
$$\tau = 10 \cdot \sum_{k=1}^{n-1} e^{-2\pi^2 \frac{\sigma^2}{\gamma^2} \cdot \frac{k}{k+1}}$$
$$\hat{c}^2 = \min \left\{ c^2 + \frac{1}{4}\gamma^2 d + \sqrt{2\log(1/\beta)} \cdot \gamma \cdot (c + \frac{1}{2}\gamma d), (c + \gamma\sqrt{d})^2 \right\}$$
$$\varepsilon = \min \left\{ \sqrt{\frac{\Delta^2 \hat{c}^2}{n\sigma^2} + 2\tau d}, \frac{\Delta\hat{c}}{\sqrt{n}\sigma} + \tau\sqrt{d} \right\}.$$

---

[9] $G$ and $H$ really consist of entries that are sums of records.

*Then $\Pi_{\mathsf{DMM}}$ satisfies $\frac{1}{2}\varepsilon^2$-concentrated differential privacy.*

*Let*

$$\hat{\sigma}^2(x) := \frac{\rho \cdot ||\boldsymbol{A}_{[T,:]}||_2^2}{d} \sum_{\tau=1}^{T} \sum_{i=1}^{n} ||\boldsymbol{g}_{\tau,i}||_2^2 + \left( \frac{\gamma^2 \cdot ||\boldsymbol{A}_{[T,:]}||_2^2}{4} + \sigma^2 \cdot ||\boldsymbol{B}_{[T,:]}||_2^2 \right) \cdot n$$

$$\leq \frac{\rho ||\boldsymbol{A}_{[T,:]}||_2^2}{d} c^2 nT + \left( \frac{\gamma^2 \cdot ||\boldsymbol{A}_{[T,:]}||_2^2}{4} + ||\boldsymbol{B}||_2^2 \cdot \sigma^2 \right) \cdot n$$

*If $\hat{\sigma}^2(x) \leq r^2$ then*

$$\mathbb{E}\left[ \left|\left| \Pi_{\mathsf{DMM}}(x) - \boldsymbol{A}_{[T,:]} \left( \sum_{i=1}^{n} \boldsymbol{x}_i \right) \right|\right|_2^2 \right] \leq \frac{dn}{1-\beta} \left( \frac{2\sqrt{2} \cdot r \cdot e^{-r^2/4\hat{\sigma}^2(x)}}{\sqrt{n(1-\beta)^{nT-1}}} \right.$$

$$\left. + \left( ||\boldsymbol{A}_{[T,:]}||_2^2 \cdot \left( \frac{\gamma^2}{4} + \frac{\beta^2 \cdot \gamma^2 n}{1-\beta} \right) + ||\boldsymbol{B}_{[T,:]}||_2^2 \cdot \sigma^2 \right)^{1/2} \right)^2.$$

We start with a modified version of Proposition 26 in (Kairouz et al., 2021a).

**Proposition B.9.** *Let $R_\gamma^G$ be as in Definition A.2 and $G = \{y \in \mathbb{R}^d : ||y||_2^2 \leq \Delta^2 \hat{c}^2\}$. Let $\Pi_{\mathsf{DMM}}'(X)$ be $\Pi_{\mathsf{DMM}}$ up to the point of modular clipping. Consider the parameters from Theorem B.8. Then $\Pi_{\mathsf{DMM}}'(X)$ satisfies $\frac{1}{2}\varepsilon^2$-concentrated differential privacy. Also the following holds.*

$$\mathbb{E}\left[ \left|\left| \Pi_{\mathsf{DMM}}'(X) - \boldsymbol{A}_{[T,:]} \sum_{i=1}^{n} \boldsymbol{X}_i \right|\right|_2^2 \right] \leq ||\boldsymbol{A}_{[T,:]}||_2^2 \cdot \left( \frac{\gamma^2 \cdot d \cdot n}{4(1-\beta)} + \left( \frac{\beta}{1-\beta} \gamma \sqrt{d} n \right)^2 \right) + ||\boldsymbol{B}_{[T,:]}||_2^2 \cdot n \cdot d \cdot \sigma^2.$$

$$\forall \boldsymbol{t} \in \mathbb{R}^d \quad \mathbb{E}\left[ \exp\left( \left\langle \boldsymbol{t}, \Pi_{\mathsf{DMM}}'(X) - \boldsymbol{A}_{[T,:]} \sum_{i=1}^{n} \boldsymbol{X}_i \right\rangle \right) \right] \leq \frac{\exp\left( \left( \frac{\gamma^2 \cdot ||\boldsymbol{A}_{[T,:]}||_2^2}{8} + \frac{\sigma^2 \cdot ||\boldsymbol{B}_{[T,:]}||_2^2}{2} \right) \cdot ||\boldsymbol{t}||_2^2 \cdot n \right)}{(1-\beta)^{nT}}.$$

*Proof.* First, the differential privacy claim follows from Kairouz et al. (2021a, Proposition 14).

Now, for the utility analysis, we have

$$\mathbb{E}\left[ \left|\left| \Pi_{\mathsf{DMM}}'(X) - \boldsymbol{A}_{[T,:]} \sum_{i=1}^{n} \boldsymbol{X}_i \right|\right|_2^2 \right] = \mathbb{E}\left[ \left|\left| \sum_{\tau=1}^{T} \boldsymbol{A}_{T,\tau} \cdot \left( \sum_{i=1}^{n} (R_\gamma^G(\boldsymbol{g}_{\tau,i}) - \boldsymbol{g}_{\tau,i}) \right) + \boldsymbol{B}_{T,\tau} \cdot \sum_{i=1}^{n} \gamma \cdot \boldsymbol{z}_{\tau,i} \right|\right|_2^2 \right]$$

$$\leq \sum_{\tau=1}^{T} \boldsymbol{A}_{T,\tau}^2 \cdot \mathbb{E}\left[ \left|\left| \sum_{i=1}^{n} R_\gamma^G(\boldsymbol{g}_{\tau,i}) - \boldsymbol{g}_{\tau,i} \right|\right|_2^2 \right] + \boldsymbol{B}_{T,\tau}^2 \cdot n \cdot \sigma^2$$

$$\leq \left|\left| \boldsymbol{A}_{[T,:]} \right|\right|_2^2 \cdot \left( \frac{\gamma^2 \cdot d \cdot n}{4(1-\beta)} + \left( \frac{\beta}{1-\beta} \gamma \sqrt{d} n \right)^2 \right) + \left|\left| \boldsymbol{B}_{[T,:]} \right|\right|_2^2 \cdot n \cdot \sigma^2,$$

where the last inequality is due directly to Proposition 26 of (Kairouz et al., 2021a).

Now, for each $i \in [n], \tau \in [T]$, we have that $R_\gamma(\boldsymbol{g}_{\tau,i}) \in \gamma \lfloor \boldsymbol{g}_{\tau,i}/\gamma \rfloor + \{0, \gamma\}^d$ and is a product distribution with mean $\boldsymbol{g}_{\tau,i}$. Thus, $R_\gamma(\boldsymbol{g}_{\tau,i}) - \boldsymbol{g}_{\tau,i} \in \{0, \gamma\}^d$ and is a product distribution with mean $\boldsymbol{0}$. Therefore, by Hoeffding's lemma, we have:

$$\forall \boldsymbol{t} \in \mathbb{R}^d \quad \mathbb{E}[\exp(\langle \boldsymbol{t}, \sum_{\tau=1}^{T} \boldsymbol{A}_{T,\tau} \sum_{i=1}^{n} R_\gamma(\boldsymbol{g}_{\tau,i}) - \boldsymbol{g}_{\tau,i} \rangle)] \leq \exp(\frac{\gamma^2}{8} \cdot n \cdot ||\boldsymbol{A}_{[T,:]}||_2^2 \cdot ||\boldsymbol{t}||_2^2).$$

Thus,

$$\forall \boldsymbol{t} \in \mathbb{R}^d \quad \mathbb{E}[\exp(\langle \boldsymbol{t}, \sum_{\tau=1}^{T} \boldsymbol{A}_{T,\tau} \sum_{i=1}^{n} R_{\gamma}^{G}(\boldsymbol{g}_{\tau,i}) - \boldsymbol{g}_{\tau,i}\rangle)] \leq \frac{\mathbb{E}[\exp(\langle \boldsymbol{t}, \sum_{\tau=1}^{T} \boldsymbol{A}_{T,\tau} \sum_{i=1}^{n} R_{\gamma}(\boldsymbol{g}_{\tau,i}) - \boldsymbol{g}_{\tau,i}\rangle)]}{\Pr[R_{\gamma}(\boldsymbol{g}_{\tau,i}) \in G \ \forall \tau, i]}$$

$$\leq \frac{\exp(\frac{\gamma^2}{8} \cdot n \cdot ||\boldsymbol{A}_{[T,:]}||_2^2 \cdot ||\boldsymbol{t}||_2^2)}{(1-\beta)^{nT}}.$$

Moreover, we have that (Canonne et al., 2020):

$$\forall \boldsymbol{t} \in \mathbb{R}^d \quad \mathbb{E}[\exp(\langle \boldsymbol{t}, \sum_{\tau=1}^{T} \boldsymbol{B}_{T,\tau} \sum_{i=1}^{n} \gamma \cdot \boldsymbol{z}_{\tau,i}\rangle)] \leq \exp(\frac{\sigma^2}{2} \cdot n \cdot ||\boldsymbol{B}_{[T:,]}||_2^2 \cdot ||\boldsymbol{t}||_2^2).$$

$\square$

Finally, we are able to prove a modified version of Theorem 36 from (Kairouz et al., 2021a).

*Proof of Theorem B.8.* First, the differential privacy follows from Proposition B.9 and the post-processing property of DP. Now, for the utility, by assumption, we have that

$$\forall \boldsymbol{x} \in \mathbb{R}^d \ \forall j \in [d] \ \forall t \in \mathbb{R} \quad \mathbb{E}[\exp(t(\boldsymbol{U}x)_j)] \leq \exp(t^2 \rho ||\boldsymbol{x}||_2^2/2d).$$

Therefore,

$$\mathbb{E}[\exp(t \cdot (\sum_{\tau=1}^{T} \boldsymbol{A}_{T,\tau} \cdot (\boldsymbol{U} \sum_{i=1}^{n} \boldsymbol{g}_{\tau,i})_j)] = \prod_{\tau=1}^{T} \cdot \prod_{i=1}^{n} \mathbb{E}[\exp(t \cdot \boldsymbol{A}_{T,\tau} \cdot (\boldsymbol{U}\boldsymbol{g}_{\tau,i})_j)]$$

$$\leq \prod_{\tau=1}^{T} \cdot \prod_{i=1}^{n} \exp(t^2 \cdot \boldsymbol{A}_{T,\tau}^2 \cdot \rho \cdot ||\boldsymbol{g}_{\tau,i}||_2^2/2d)$$

$$= \exp(t^2 \cdot ||\boldsymbol{A}_{[T,:]}||_2^2 \cdot \rho \cdot \sum_{\tau=1}^{T} \sum_{i=1}^{n} ||\boldsymbol{g}_{\tau,i}||_2^2/2d).$$

Combining with the result of Proposition B.9, we have

$$\forall t \in \mathbb{R} \ \forall j \in [d] \quad \mathbb{E}[\exp(t \cdot (\mathcal{A}(\boldsymbol{U}x))_j)] \leq \exp(\frac{t^2 \cdot ||\boldsymbol{A}_{[T,:]}||_2^2 \cdot \rho}{2d} \cdot \sum_{\tau=1}^{T} \sum_{i=1}^{n} ||\boldsymbol{g}_{\tau,i}||_2^2)$$

$$\cdot \frac{\exp((\frac{\gamma^2 \cdot ||\boldsymbol{A}_{[T,:]}||_2^2}{8} + \frac{\sigma^2 \cdot ||\boldsymbol{B}_{[T,:]}||_2^2}{2}) \cdot t^2 \cdot n)}{(1-\beta)^{nT}}$$

Recall $\hat{\sigma}^2(x) = \frac{\rho \cdot ||\boldsymbol{A}_{[T,:]}||_2^2}{d} \sum_{\tau=1}^{T} \sum_{i=1}^{n} ||\boldsymbol{g}_{\tau,i}||_2^2 + (\frac{\gamma^2 \cdot ||\boldsymbol{A}_{[T,:]}||_2^2}{4} + \sigma^2 \cdot ||\boldsymbol{B}_{[T,:]}||_2^2) \cdot n$.

By Proposition 35 of (Kairouz et al., 2021a), for all $j \in [d]$,

$$\mathbb{E}[(M_{[a,b]}(\Pi_{\text{DMM}}'(\boldsymbol{U}x))_j - \Pi_{\text{DMM}}'(\boldsymbol{U}x)_j)^2] \leq (b-a)^2 \cdot \frac{1}{(1-\beta)^{nT}} \cdot e^{-(b-a)^2/8\hat{\sigma}^2(x)} \cdot (e^{\frac{a^2 - b^2}{4\hat{\sigma}^2}} + e^{\frac{b^2 - a^2}{4\sigma^2}}),$$

where $a = -r$ and $b = r$ here. Summing over $j \in [d]$ gives

$$\mathbb{E}[||M_{[-r,r]}(\Pi_{\text{DMM}}'(\boldsymbol{U}x)) - \Pi_{\text{DMM}}'(\boldsymbol{U}x)||_2^2] \leq 4r^2 \cdot \frac{d}{(1-\beta)^{nT}} \cdot e^{-r^2/2\hat{\sigma}^2(x)} \cdot 2$$

Continuing with the proof from (Kairouz et al., 2021a), we get:

$$\mathbb{E}[||\Pi_{\mathsf{DMM}}(x) - \boldsymbol{A}_{[T,:]} \sum_{i=1} \boldsymbol{X}_i||_2^2]$$

$$\leq \left( \left(8r^2 \cdot \frac{d}{(1-\beta)^{nT}} \cdot e^{-r^2/2\hat{\sigma}^2(x)}\right)^{1/2} + \left(||\boldsymbol{A}_{[T,:]}||_2^2 \cdot \left(\frac{\gamma^2 \cdot d \cdot n}{4(1-\beta)} + \left(\frac{\beta}{1-\beta}\gamma\sqrt{d}n\right)^2\right) + \right. \right.$$

$$\left. \left. ||\boldsymbol{B}_{[T,:]}||_2^2 \cdot n \cdot d \cdot \sigma^2\right)^{1/2}\right)^2$$

$$= \frac{dn}{1-\beta}\left(\frac{2\sqrt{2} \cdot r \cdot e^{-r^2/4\hat{\sigma}^2(x)}}{\sqrt{n(1-\beta)^{nT-1}}} + \left(||\boldsymbol{A}_{[T,:]}||_2^2 \cdot \left(\frac{\gamma^2}{4} + \frac{\beta^2 \cdot \gamma^2 n}{1-\beta}\right) + ||\boldsymbol{B}_{[T,:]}||_2^2 \cdot \sigma^2\right)^{1/2}\right)^2.$$

$\square$

With this error bound, assuming that $\beta \leq 1/\sqrt{n}$ and $\hat{\sigma}^2(x) \leq r^2/4\log(r\sqrt{n}/\gamma^2)$, we get

$$\mathbb{E}[||\tilde{\mathcal{A}}(x) - \boldsymbol{A}_{[T,:]} \sum_{i=1} \boldsymbol{X}_i||_2^2] \leq O(dn((||\boldsymbol{A}_{[T,:]}||_2^2 \cdot \gamma^2 + ||\boldsymbol{B}_{[T,:]}||_2^2 \cdot \sigma^2)).$$

*Proof of Theorem 4.2.* Note that $r = \frac{1}{2}\gamma m$. We verify that setting the parameters as specified yields $\frac{1}{2}\varepsilon^2$-concentrated DP and the desired accuracy. First, we have that

$$\varepsilon^2 \leq \frac{\Delta^2\hat{c}^2}{n\sigma^2} + 2\tau d \leq \frac{\Delta^2(c+\gamma\sqrt{d})^2}{n\sigma^2} + 20nde^{-\pi^2(\sigma/\gamma)^2} \leq \frac{2\Delta^2c^2}{n\sigma^2} + \frac{2d\Delta^2}{n(\sigma/\gamma)^2} + 20nde^{-\pi^2(\sigma/\gamma)^2}.$$

Thus the privacy requirement is satisfied as long as $\sigma \geq 2c\Delta/\varepsilon\sqrt{n}$ and $(\sigma/\gamma)^2 \geq 8d\Delta^2/\varepsilon^2 n$, and $20nde^{-\pi^2(\sigma/\gamma)^2} \leq \varepsilon^2/4$. So we can set

$$\sigma = \max\left\{\frac{2c\Delta}{\varepsilon\sqrt{n}}, \frac{\gamma\Delta\sqrt{8d}}{\varepsilon\sqrt{n}}, \frac{\gamma}{\pi^2}\log(\frac{80nd}{\varepsilon^2})\right\} = \tilde{\Theta}(\frac{c\Delta}{\varepsilon\sqrt{n}} + \sqrt{\frac{d}{n}} \cdot \frac{\gamma\Delta}{\varepsilon} + \gamma\log(\frac{nd}{\varepsilon^2})).$$

We set $\beta = \min\{1/n, 1/2\} = \Theta(\frac{1}{n})$.

Next,

$$\hat{\sigma}^2 \leq \frac{\rho||\boldsymbol{A}_{[T,:]}||_2^2}{d}c^2nT + (\frac{\gamma^2||\boldsymbol{A}_{[T,:]}||_2^2}{4} + \sigma^2||\boldsymbol{B}_{[T,:]}||_2^2) \cdot n$$

$$\leq \frac{\rho||\boldsymbol{A}_{[T,:]}||_2^2}{d}c^2nT + \gamma^2||\boldsymbol{A}_{[T,:]}||_2^2n + \sigma^2||\boldsymbol{B}_{[T,:]}||_2^2 \cdot n$$

$$\leq O(\frac{\rho||\boldsymbol{A}_{[T,:]}||_2^2}{d}c^2nT + \gamma^2||\boldsymbol{A}_{[T,:]}||_2^2n + ||\boldsymbol{B}_{[T,:]}||_2^2(\frac{c^2\Delta^2}{\varepsilon^2} + \frac{\gamma^2d\Delta}{\varepsilon^2} + \gamma^2n\log^2(\frac{nd}{\varepsilon^2})))$$

$$\leq O(\frac{\rho||\boldsymbol{A}_{[T,:]}||_2^2}{d}c^2nT + ||\boldsymbol{B}_{[T,:]}||_2^2\frac{c^2\Delta^2}{\varepsilon^2})) + \gamma^2 \cdot O(||\boldsymbol{A}_{[T,:]}||_2^2n + ||\boldsymbol{B}_{[T,:]}||_2^2(\frac{d\Delta}{\varepsilon^2} + n\log^2(\frac{nd}{\varepsilon^2})).$$

Now we work out the asymptotics of the accuracy guarantee:

$$\mathbb{E}[||\Pi_{\mathsf{DMM}}(X) - \boldsymbol{A}_{[T,:]}\sum_{i=1}\boldsymbol{X}_i||_2^2]$$

$$\leq \frac{dn}{1-\beta}\left(\frac{2\sqrt{2}\cdot r\cdot e^{-r^2/4\hat{\sigma}^2(x)}}{\sqrt{n(1-\beta)^{nT-1}}} + \left(||\boldsymbol{A}_{[T,:]}||_2^2\cdot\left(\frac{\gamma^2}{4}+\frac{\beta^2\cdot\gamma^2 n}{1-\beta}\right) + ||\boldsymbol{B}_{[T,:]}||_2^2\cdot\sigma^2\right)^{1/2}\right)^2.$$

$$\leq O(nd(\frac{re^{-r^2/4\hat{\sigma}^2}}{\sqrt{n}} + \sqrt{||\boldsymbol{A}_{[T,:]}||_2^2\gamma^2 + ||\boldsymbol{B}_{[T,:]}||_2^2\sigma^2}))$$

$$\leq O(nd(\frac{r^2 e^{-r^2/2\hat{\sigma}^2}}{n} + ||\boldsymbol{A}_{[T,:]}||_2^2\gamma^2 + ||\boldsymbol{B}_{[T,:]}||_2^2\sigma^2))$$

$$\leq O(nd(\frac{\gamma^2 m^2}{n}\exp(\frac{-\gamma^2 m^2}{8\hat{\sigma}^2}) + ||\boldsymbol{A}_{[T,:]}||_2^2\gamma^2 + ||\boldsymbol{B}_{[T,:]}||_2^2(\frac{c^2\Delta^2}{\varepsilon^2 n} + \frac{d\gamma^2\Delta^2}{\varepsilon^2 n} + \gamma^2\log^2(\frac{nd}{\varepsilon^2}))))$$

$$\leq O(||\boldsymbol{B}_{[T,:]}||_2^2\frac{c^2\Delta^2 d}{\varepsilon^2} + \gamma^2 nd(\frac{m^2}{n}\exp(\frac{-\gamma^2 m^2}{8\hat{\sigma}^2}) + ||\boldsymbol{A}_{[T,:]}||_2^2 + ||\boldsymbol{B}_{[T,:]}||_2^2(\frac{d\Delta^2}{\varepsilon^2 n} + \log^2(\frac{nd}{\varepsilon^2}))))$$

Similarly to the analysis of Theorem 2 in (Kairouz et al., 2021a), if

$$m^2 \geq O((||\boldsymbol{A}_{[\boldsymbol{T},:]}||_2^2 n + ||\boldsymbol{B}_{[\boldsymbol{T},:]}||_2^2(\frac{d\Delta}{\varepsilon^2} + n\log^2(\frac{nd}{\varepsilon^2})))\cdot\log(1+m^2/n)$$

$$= \tilde{O}(||\boldsymbol{A}_{[\boldsymbol{T},:]}||_2^2 n + ||\boldsymbol{B}_{[\boldsymbol{T},:]}||_2^2(\frac{d\Delta}{\varepsilon^2} + n)),$$

then we can set

$$\gamma^2 = O(\frac{\rho||\boldsymbol{A}_{[T,:]}||_2^2 c^2 nT}{d} + \frac{||\boldsymbol{B}_{[T,:]}||_2^2 c^2\Delta^2}{\varepsilon^2})\cdot\frac{\log(1+m^2/n)}{m^2}$$

so that $\frac{m^2}{n}\exp(\frac{-\gamma^2 m^2{}^2}{8\hat{\sigma}}) \leq 1$.

This gives us,

$$\mathbb{E}[||\tilde{\mathcal{A}}(x) - \boldsymbol{A}_{[T,:]}\sum_{i=1}\boldsymbol{X}_i||_2^2]$$

$$\leq O(||\boldsymbol{B}_{[T,:]}||_2^2\frac{c^2\Delta^2 d}{\varepsilon^2} + \gamma^2 nd(1 + ||\boldsymbol{A}_{[T,:]}||_2^2 + ||\boldsymbol{B}_{[T,:]}||_2^2(\frac{d\Delta^2}{\varepsilon^2 n} + \log^2(\frac{nd}{\varepsilon^2}))))$$

$$\leq O(||\boldsymbol{B}_{[T,:]}||_2^2\frac{c^2\Delta^2 d}{\varepsilon^2} + (\frac{\rho||\boldsymbol{A}_{[T,:]}||_2^2 c^2 nT}{d} + \frac{||\boldsymbol{B}_{[T,:]}||_2^2 c^2\Delta^2}{\varepsilon^2})$$

$$\cdot\frac{\log(1+m^2/n)}{m^2}nd(1 + ||\boldsymbol{A}_{[T,:]}||_2^2 + ||\boldsymbol{B}_{[T,:]}||_2^2(\frac{d\Delta^2}{\varepsilon^2 n} + \log^2(\frac{nd}{\varepsilon^2}))))$$

$$\leq O(||\boldsymbol{B}_{[T,:]}||_2^2\frac{c^2\Delta^2 d}{\varepsilon^2} + ||\boldsymbol{B}_{[T,:]}||_2^2\frac{c^2\Delta^2 d}{\varepsilon^2}(\frac{\log(1+m^2/n)}{m^2}n$$

$$\cdot(\rho||\boldsymbol{A}_{[T,:]}||_2^2 T + 1 + ||\boldsymbol{A}_{[T,:]}||_2^2 + ||\boldsymbol{B}_{[T,:]}||_2^2(\frac{d\Delta^2}{\varepsilon^2 n} + \log^2(\frac{nd}{\varepsilon^2})))))$$

$$\leq O(||\boldsymbol{B}_{[T,:]}||_2^2\frac{c^2\Delta^2 d}{\varepsilon^2}(1 + \frac{\log(1+m^2/n)}{m^2}n\cdot(\rho||\boldsymbol{A}_{[T,:]}||_2^2 T + 1 + ||\boldsymbol{A}_{[T,:]}||_2^2 + ||\boldsymbol{B}_{[T,:]}||_2^2(\frac{d\Delta^2}{\varepsilon^2 n} + \log^2(\frac{nd}{\varepsilon^2}))))).$$

So, if

$$m^2 \geq O(\log(1+m^2/n)n\cdot(\rho||\boldsymbol{A}_{[T,:]}||_2^2 T + 1 + ||\boldsymbol{A}_{[T,:]}||_2^2 + ||\boldsymbol{B}_{[T,:]}||_2^2(\frac{d\Delta^2}{\varepsilon^2 n} + \log^2(\frac{nd}{\varepsilon^2}))))$$

$$= \tilde{O}(\rho||\boldsymbol{A}_{[T,:]}||_2^2 nT + ||\boldsymbol{B}_{[T,:]}||_2^2\frac{d\Delta^2}{\varepsilon^2}),$$

then the mean squared error is $O(||\boldsymbol{B}_{[T,:]}||_2^2\frac{c^2\Delta^2 d}{\varepsilon^2})$, as required. The final bound is obtained by simply summing the above over each round from $T=1$ to $T=T^*$. $\qquad\square$

# C. DMM Security Model and Proof

## C.1. Security proofs

We first provide an intuition on the current analysis for proving the security of cryptographic protocols. In the security proof, we compare between an $n$-party function $f(x_1, \ldots, x_n) = (y_1, \ldots, y_n)$ and a protocol $P(x_1, \ldots, x_n)$ that allegedly privately computes the function $f$. Intuitively, a protocol $P$ correctly and privately computes $f$ if the following hold: (a) *Correctness:* For every input $\vec{x} = (x_1, \ldots, x_n)$, the output of the parties at the end of the protocol interaction $P$ is the same as $f(\vec{x})$; (b) *Privacy*: There exists a simulator $\mathcal{S}$ that receives the input and output of the corrupted parties, and can efficiently generate the messages that the corrupted parties received during the protocol execution. The simulator does not know the input/outputs of the honest parties. Intuitively, the fact that the messages sent by the honest parties can be generated from the input/output of the corrupted parties implies that these messages do not contain any additional information about the inputs of the honest parties besides what is revealed from the output of the computation.

## C.2. Security Model

We now introduce the formal security model. We first note that we consider robustness checks on inputs out of the scope of our security model; i.e., we do not cover *poisoning attacks*, which have been extensively studied in the literature, e.g., (Tolpegin et al., 2020; Fang et al., 2020). Indeed, it is the case that malicious parties can input to the protocol whatever they want as their gradients and noise $\boldsymbol{g}, \boldsymbol{z}$, which can lead to a meaningless model.

We follow the standard real/ideal world security paradigm of (Goldreich, 2004). Consider some multi-party protocol $\Pi$ that is executed by some parties $P_1, \ldots, P_N$ that are grouped into committees $\mathcal{C}_1, \ldots, \mathcal{C}_{T^*}$ from iteration 1 to iteration $T^*$ and a server $S$. Note: the committees can be arbitrarily chosen, but our protocol only provides security if the assumption that the number of parties $\mathcal{A}$ corrupts is at most $t_c$ holds; in other words, we abstract out the committee selection process.[10] Each of these parties has inputs $\boldsymbol{x}_1, \ldots, \boldsymbol{x}_N$, and they want to evaluate some given *functionality* $\mathcal{F}$. In our case, the functionality $\mathcal{F}_{\mathsf{PPFL}}$ is resharing the inputs from all previous committees to the next committee, in each iteration, and then outputting the $\widehat{\boldsymbol{AX}}_T$ value to the sever in each iteration $T$, given some factorization $\boldsymbol{A} = \boldsymbol{BC}$. The security of protocol $\Pi$ is defined by comparing the real-world execution of the protocol with an *ideal*-world evaluation of $\mathcal{F}$ by a trusted party (ideal functionality), who receives the inputs $\boldsymbol{x}_1, \ldots, \boldsymbol{x}_N$ from the parties in the clear and simply sends the relevant parties their outputs $\mathcal{F}(\boldsymbol{x}_1, \ldots, \boldsymbol{x}_N)$ periodically. There is an adversary $\mathcal{A}$ that chooses to corrupt at most $t_c$ of the $n$ parties in each iteration, along with the server. This adversary $\mathcal{A}$ sees all of the messages and inputs and outputs of the corrupted parties and is allowed to act arbitrarily on their behalf. Informally, it is required that for every adversary that corrupts some parties during the protocol execution, there is an adversary $\mathcal{S}$, also referred to as the *simulator*, which can achieve the same effect and learn the same information in the ideal-world. This simulator only sees what the corrupted parties send to the honest parties and the outputs, not the inputs $\boldsymbol{x}$ of the honest parties. We now formally describe the security definition.

**Real Execution.** In the real execution, $\Pi$ is executed in the presence of the adversary $\mathcal{A}$. The *view* of a party $P$ during an execution of $\Pi$, denoted by $\mathsf{View}_P^\Pi$ consists of the messages $P$ receives from the other parties during the execution and $P$'s input. The execution of $\Pi$ in the presence of $\mathcal{A}$ on inputs $(\boldsymbol{x}_1, \ldots, \boldsymbol{x}_N)$ denoted $\mathsf{Real}_{\Pi,\mathcal{A}}(\boldsymbol{x}_1, \ldots, \boldsymbol{x}_N)$ is defined as $\{\mathsf{View}_P^\Pi\}_{P \in \mathfrak{C}}$. The output of $\Pi$ to the honest parties in the presence of $\mathcal{A}$ on inputs $(\boldsymbol{x}_1, \ldots, \boldsymbol{x}_N)$ is noted as $\mathsf{Output}$.

**Ideal Execution.** In the ideal execution, the parties and an ideal world adversary $\mathcal{S}$ interact with a trusted party (ideal functionality). The ideal execution proceeds as follows: As a committee $\mathcal{C}_T$ comes online, the parties $P_{T,1}, \ldots, P_{T,n}$ in that committee send their inputs $\boldsymbol{x}_{T,1}, \ldots, \boldsymbol{x}_{T,n}$ to the trusted party, who computes the output $\mathcal{F}(\boldsymbol{x}_{1,1}, \ldots, \boldsymbol{x}_{T,n})$ to the server for that iteration. $\mathcal{S}$ is also allowed to release a vector $\boldsymbol{\chi}$, which will be added to the output, to simulate additive attacks.

**Definition C.1.** Protocol $\Pi$ securely computes $\mathcal{F}$ if for every adversary $\mathcal{A}$ there exists a simulator $\mathcal{S}$ such that

$$\mathsf{SD}((\{\mathsf{View}_P^\Pi\}_{P \in \mathfrak{C}}, \mathsf{Output}), (\mathcal{S}(\{\boldsymbol{x}_{T^*,j}\}_{T,j \in \mathfrak{C}(T)}, \mathcal{F}(\boldsymbol{x}_{1,1}, \ldots, \boldsymbol{x}_{T^*,n}), \mathcal{F}(\boldsymbol{x}_{1,1}, \ldots, \boldsymbol{x}_{T^*,n}) + \boldsymbol{\chi})) \leq \mathsf{negl}(\lambda), \text{[11]}$$

where $\mathsf{SD}$ is the statistical distance between the two distributions, $\mathfrak{C}(T)$ is the set of corrupted parties in iteration $T$, and $\lambda$ is the *security parameter*.

---

[10] In practice, the committee selection is done by the server.

[11] $\mathsf{negl}(\lambda)$ is any function in $\lambda^{\omega(1)}$

## C.3. Security Proof

We now give the formal security proof.

**Theorem C.2** (Security). $\Pi_{\mathsf{DMM}}$ *securely computes* $\mathcal{F}_{\mathsf{PPFL}}$ *for* $t_c + t_d < (1/2 - \mu)n$, $0 < \mu < 1/2$.

*Proof.* We first build the simulator $\mathcal{S}$. The simulator runs $\mathcal{A}$ internally. We describe the simulator for the first iteration $T = 1$ and then inductively for the rest. Throughout, we will (inductively) show that the simulator knows all of the corrupted parties' shares. We start with the case of a corrupted server $S$.

**Corrupted Server** In iteration 1, $\mathcal{S}$ simulates the shares sent by honest parties of iteration 1 to corrupted parties of iteration 1 and 2 by sampling random values from the field $\mathbb{F}$. $\mathcal{S}$ receives on behalf of the honest parties in committee $\mathcal{C}_1$ the shares sent by corrupted parties from the same committee. Note that the honest shares completely (and exactly) define these sharings since the number of honest parties is exactly $t_c + k$, and thus $\mathcal{S}$ can compute the corrupted parties' shares and the underlying secret $\boldsymbol{y}_i$. With the corrupted parties' shares, along with the output for iteration $T = 1$ (which $\mathcal{S}$ receives since the server is corrupted), $\mathcal{S}$ has $t_c + k$ points on the $(t_c + k - 1)$-degree polynomial underlying each output $[\boldsymbol{Y}_{1,\ell}]$, and thus can reconstruct the whole sharing. Based on this, the simulator can send the honest parties' shares to the server.

In subsequent iterations $T > 1$, $\mathcal{S}$ first simulates the shares of honest parties' new gradients and noise as above, along with the reshares to the next committee. It also receives the corrupted parties' input shares from iteration $T$, and can reconstruct the whole sharing $[\boldsymbol{y}_i]$, in the same way as above. $\mathcal{S}$ also receives on behalf of the honest parties in committee $\mathcal{C}_T$ the reshared shares sent by corrupted parties from iteration $T - 1$. Note that again the honest shares completely (and exactly) define these sharings since the number of honest parties is exactly $t_c + k$, and thus $\mathcal{S}$ can compute the corrupted parties' shares as well as the actual underlying reshared shares $\tilde{\boldsymbol{Z}}_1^i, \ldots, \tilde{\boldsymbol{Z}}_k^i$ of each corrupted party $P_i$ in $\mathcal{C}_T$. Note that these might be different from the actual underlying shares $\hat{\boldsymbol{Z}}_1^i, \ldots, \hat{\boldsymbol{Z}}_k^i$ of the corrupted parties which, $\mathcal{S}$ knows from above. Thus, $\mathcal{S}$ can compute $\boldsymbol{e}_m^i \leftarrow \hat{\boldsymbol{Z}}_m^i - \tilde{\boldsymbol{Z}}_m^i$ for each $m \in [k]$. Since reconstruction used within Recover of LRP is just a constant $\lambda_i^j$ multiplied by a party's share, the error introduced here is $\lambda_i^j \cdot \boldsymbol{e}_m^i$ for each $i \in \mathfrak{C}(T)$. Thus $\mathcal{S}$ sets $\boldsymbol{\chi}_T \leftarrow \sum_{i \in \mathfrak{C}(1)} \lambda_i^j \cdot \boldsymbol{e}_m^i$. This error will be incorporated in honest parties' shares of $\widehat{\boldsymbol{A}\boldsymbol{X}}$ for all iterations including and after $T$. Indeed, in iteration $T$, $\mathcal{S}$ uses the computed corrupted parties' shares, along with the output for iteration $T$ with the error $\boldsymbol{\chi}_\tau$ for all previous iterations $\tau \leq T$ added in, to obtain $t_c + k$ values with which it can reconstruct the whole sharing as above, and thus the shares that honest parties send to the server.

Now we show that this is a good simulation. By the properties of Shamir Secret Sharing, we know that the at most $t_c$ shares that the adversary receives in the real world for every sharing will be distributed randomly. Thus the shares that $\mathcal{S}$ sends are distributed same way. We also showed that the errors $\boldsymbol{\chi}_T$ and the shares of the corrupted parties are computed exactly as in the real world. Therefore $\mathcal{S}$ perfectly simulates the real world.[12]

**Honest Server** In the case of an honest server, we can use all of the same simulation above, except instead of simulating the output shares to the server, we can use this same simulation to compute the error values $\boldsymbol{\chi}_T$ added to the honest output. We also need to show that, even in the presence of honest dropout parties, if the corrupted parties do not deviate from the protocol description, then the correct values are output to the server. This is true since the number of honest parties that do not dropout is at least $n - t_d > (1/2 + \mu)n$ and $k \leq 2\mu n$. Indeed, $t_c + k \leq (1/2 + \mu)n < n - t_d$, so the shares of the parties that do not dropout can still be used to reconstruct the secrets, both during Recover of LRP and the actual output reconstruction to the server in each iteration.

This completes the security proof. $\qquad\square$

## D. Additional Experimental Results

**FEMNIST details.** Federated EMNIST is an image classification dataset containing 671,585 training handwritten digit/letter images over 64 classes grouped into $N = 3400$ clients by their writer. We use the standard dataset split provided by TensorFlow. As in (Kairouz et al., 2021a), we train a small convolutional net with two $3 \times 3$ conv layers with $32/64$ channels followed by two fully connected layers with $128/62$ output units; a $2 \times 2$ max pooling layer and two dropout layers with

---

[12]Note that both in the real and ideal world, if an honest party ever hears from less than $t_c + k$ parties from the previous iteration, they will abort since then more parties have dropped out (or have been dropped by the server) than can be tolerated.

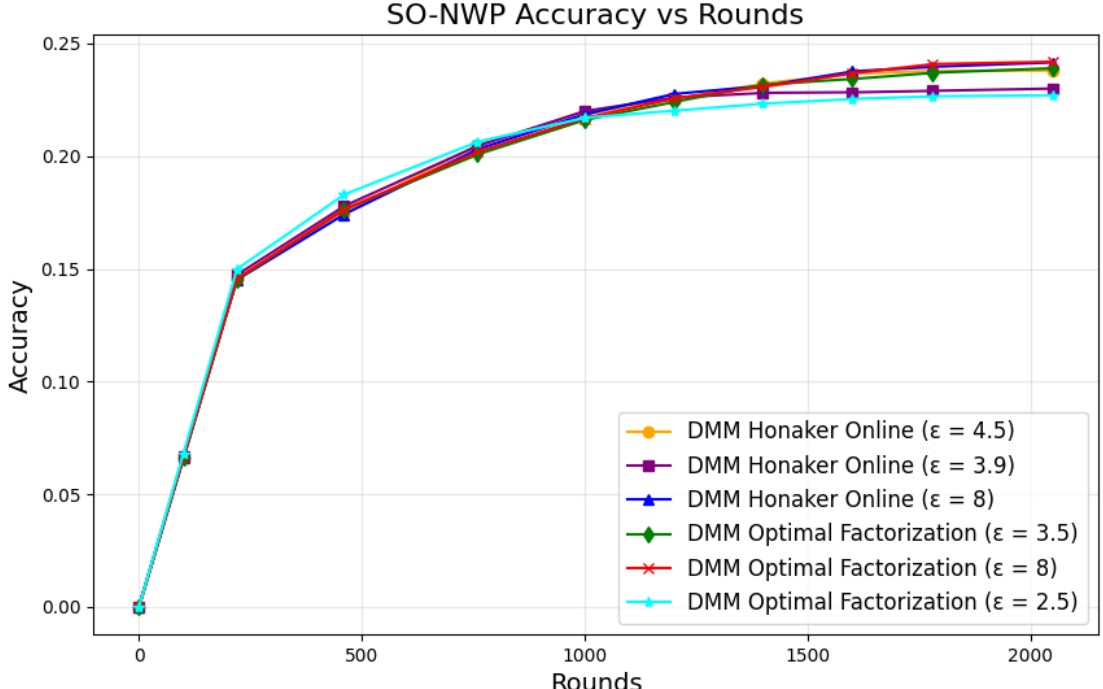

*Figure 5.* Accuracies during training for SO-NWP across different $\varepsilon$ for both the optimal factorization and Honaker Online facorization

drop rate $0.25/0.5$ are added after the first 3 trainable layers, respectively. The total number of parameters is $d = 1018174$. We use namely momentum $0.9$, 1 client training epoch per iteration, client learning rate $\eta_c = 0.02$, server learning rate $\eta_s = 1$, and client batch size to 16.

**SO-NWP details.** Stack Overflow is a large-scale text dataset based on the question answering site Stack Overflow. It contains over 108 training sentences extracted from the site grouped by the $N = 342477$ users, and each sentence has associated metadata such as tags. The task of SO-NWP involves predicting the next words given the preceding words in a sentence We use the standard dataset split provided by TensorFlow. As in (Kairouz et al., 2021a; Choquette-Choo et al., 2023b), we use the LSTM architecture defined in (Reddi et al., 2021) directly, which has a model size of $d = 4050748$ parameters (slightly under $2^{22}$). We use namely momentum $0.9$, 1 client training epoch per iteration, client learning rate $\eta_c = 0.02$, server learning rate $\eta_s = 1$, and client batch size to 16.

**Accuracy Across Training iterations.** In Figures 5 and 6, we show how the accuracies of our different models vary across training iterations. We show the results for the different matrix factorizations (Honaker Online and optimal) and different privacy values $\varepsilon$.

**Privacy Guarantees with Dropouts and Corrupted Parties.** We note that, just as in (Kairouz et al., 2021a), our privacy guarantees degrade with corrupt parties and honest dropouts—the amount of combined noise in each iteration is proportional to $(1 - \mu')n$ instead of $n$, where $\mu'$ is the fraction of parties that are corrupted or dropped out (recall that we assume $\mu' < (1/2 - \mu)$, for $0 < \mu < 1/2$). Indeed, the actual obtained $\varepsilon'$ value for DP scales the originally derived $\varepsilon$ value by a $\approx 1/(1 - \mu')$ factor. See Kairouz et al. (2021a, Figure 9) for a graphical representation.

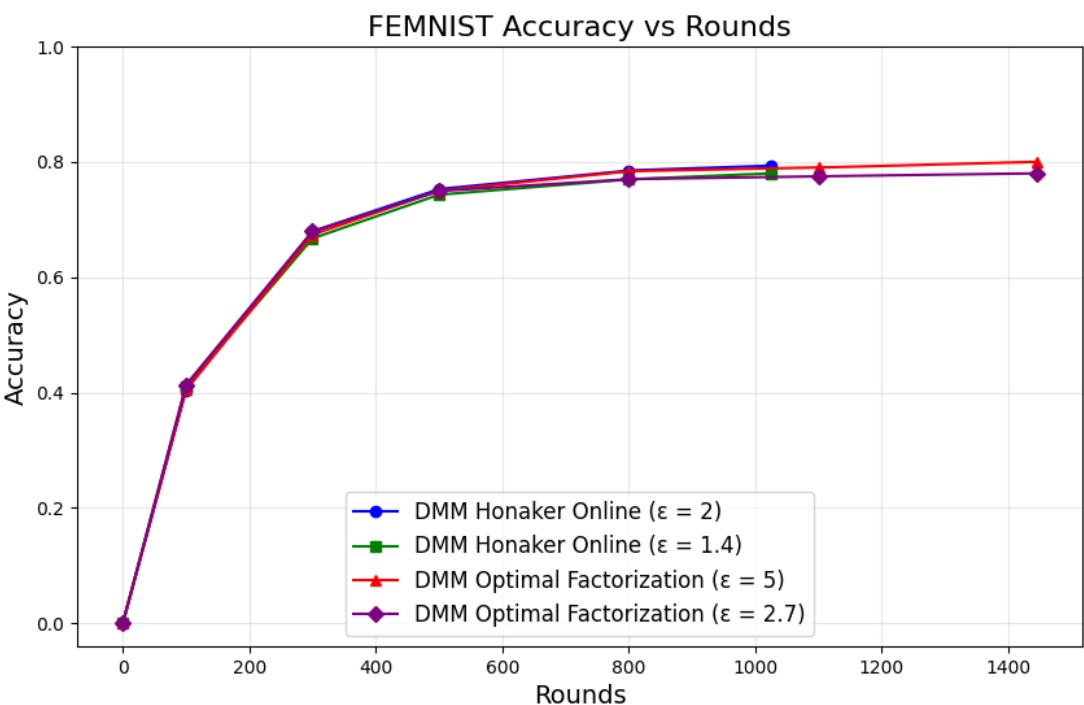

*Figure 6.* Accuracies during training for FEMNIST across different $\varepsilon$ for both the optimal factorization and Honaker Online facorization

