# OpenReview forum: "DMM: Distributed Matrix Mechanism for Differentially-Private Federated Learning Based on Constant-Overhead Linear Secret Resharing"
_ICML.cc/2025/Conference — ICML 2025 poster_

### Official Review · Reviewer_M2TL · 2025-03-09

**Overall Recommendation:** 4

**Summary:**

This paper introduces a distributed matrix mechanism for federated learning. It is a local DP mechanism where each client of FL uses their own noises instead of the server adds the noise. To achieve this goal, the authors use cryptographic protocols for secrete sharing to achieve this. The proposed frame is robust, as it could handle corrupted clients and dropped out honest clients (as long as they are not too many for the problem to be not recoverable). Experiments are conducted to show that the proposed method has good tradeoff between $\epsilon$ and the accuracy compared to prior local DP method for FL, and it dramatically saves the communication cost compared to implement the crypto protocol naively.

## Update after rebuttal
Authors explained the reason for using discrete noise is due to the use of cryptography tools that work only for finite field. They are encouraged to incorporate this rationale into the revision. As my score was on the positive side, I'll keep it as is.

**Claims And Evidence:**

Yes, the statements made in the paper are backed up by proofs, and extensive experiments are performed to show the good accuracy of proposed method against other local DP methods, and the communication cost saving.

**Essential References Not Discussed:**

The paper cites proper references.

**Experimental Designs Or Analyses:**

The experimental design and analyses are quite standard.

**Methods And Evaluation Criteria:**

Yes, datasets used are standard and comparisons are fair.

**Other Comments Or Suggestions:**

Some suggestions:

1. The paper is very technically dense due to combining tools from different fields, therefore certain parts are hard to parse (for me, since I'm more familiar with DP and FL, the secret sharing part is quite hard to read). It might be helpful to start with an overview of the techniques so that each part is packaged in a more black-box fashion (e.g., abstract the secret sharing, or abstract the matrix mechanism) so that readers can get a better sense on the high level details before diving into heavily technical parts.

2. The paper could benefit from providing a table comparing the error guarantees against other local DP method, and central matrix mechanism (maybe in appendix). It would give a better sense on various tradeoffs for local v.s. central, non-matrix mechanism v.s. matrix mechanism.

**Other Strengths And Weaknesses:**

Overall, I think the paper presents interesting results and experiments are performed to show the validity of their proposed method.

**Questions For Authors:**

Why were discrete Gaussian noises used? Is it for the ease of practical implementations?

**Relation To Broader Scientific Literature:**

I think the contributions of this paper are significant. Local DP for FL is an important subject of study as it does not require server to access noise and can just do simple average. This paper in particular studies matrix mechanism, a powerful DP tool that achieves good privacy and utility. To make it work, it introduces novel secret sharing into the picture to further reduce communication cost. This combination of secret sharing and matrix mechanism is interesting, and I believe publishing this paper would be beneficial to the FL community.

**Theoretical Claims:**

I read the DP proofs, I think they are correct. However I'm not familiar with cryptography.

---

> ### Author Rebuttal · Authors · 2025-04-01
>
> 1) Why discrete Gaussians?:
>
> Since our novel techniques are independent of the noise distribution used, we just chose one distribution that we believe to be the most well-studied/understood: the discrete Gaussian distribution. Indeed, we just need CX + Z to be differentially-private, where Z can be sampled from any (discrete) noise distribution; whichever distribution yields the best results can be used and these results will be at least as good as those stated in our paper. We are not currently aware of any other (discrete) noise distribution that is always better than the discrete Gaussian. Ultimately, the main message of our paper is the same: using the matrix mechanism for local (distributed) DP is better than using uncorrelated noise for local (distributed) DP.

---

> > ### Comment · Reviewer_M2TL · 2025-04-02
> >
> > I thank authors for the response, I guess I didn't state my question clearly: why would you use a discrete noise distribution? Can you simply use Gaussian distribution? Is using a discrete noise distribution due to practical consideration, or does it offer certain theoretical advantages?

---

> > > ### Author Response · Authors · 2025-04-02
> > >
> > > There are two main reasons why we use a discrete noise distribution: (1) Cryptographic techniques like secret sharing and Secure Aggregation work over finite, discrete algebraic structures (like finite fields), and thus discrete noise should be used; (2) Computers also can only represent a finite set of numbers in a discrete, noncontinuous fashion in reality and it has been shown that attempting to use the continuous Gaussian in concrete implementations breaks privacy guarantees (Mironov, 2012: “On significance of the least significant bits for differential privacy”).
> > >
> > > Many previous DP papers have used discrete noise distributions for this reason, including (Kairouz et al., 2021a), to whom we compare, and (Agarwal et al., 2021).

---

### Official Review · Reviewer_sHuj · 2025-03-13

**Overall Recommendation:** 3

**Summary:**

The paper proposes a Linear Secrete Sharing Protocol with a Constant-Overhead Linear Secret Resharing method to facilitate matrix mechanism to preserve DP in Federated Learning. In the case of Distributed FL, in which the clients do not trust the server, the clients will generate the noise and share to each other their gradients. The proposed method facilitates this process with marginal communication overhead.

**Claims And Evidence:**

Yes

**Essential References Not Discussed:**

Not Applicable

**Experimental Designs Or Analyses:**

I did check the validity of the experiments in the main body.
- Figure 6, How can eps=5 and eps=2.7 be run at the same epoch and achieve the same accuracy? To achieve eps=2.7 with the same # of rounds, you need a larger noise scale leading to bigger degradation of performance

**Methods And Evaluation Criteria:**

Yes

**Other Comments Or Suggestions:**

- It would be better to discuss a practical setting before diving into the mathematical notation.
- The notations are hard to follow.

**Other Strengths And Weaknesses:**

Strength:
- Thorough theoretical and experimental analysis.
- Adapt existing works with modification, which is easy to adopt.

Weakness:
- Lack of practical setting and motivation.
- The theoretical analysis heavily depends on existing works.

**Questions For Authors:**

- In the case of distributed FL, how can matrix A be constructed? Since the learning process does not include a hosting server to construct this matrix, how can each user can construct this?
- Figure 6, How can eps=5 and eps=2.7 be run at the same epoch and achieve the same accuracy? To achieve eps=2.7 with the same # of rounds, you need a larger noise scale leading to bigger degradation of performance

**Relation To Broader Scientific Literature:**

Not Applicable

**Theoretical Claims:**

I did check the correctness of all the theoretical proofs.
- Theorem 1.
- Theorem 2.

I don't see any issues with these theorems.

---

> ### Author Rebuttal · Authors · 2025-04-01
>
> 1) Construction of matrix A:
>
> The matrix A (and its factorization A=BC) is a public value that can be constructed before training begins by the central server (which does still exist in our framework, though importantly privacy holds even with respect to this server; see, e.g., right of Figure 1).
>
> 2) Figure 6, performance of eps=5 and eps=2.7 at given round:
>
> Note that it is common for the accuracy at different privacy levels to be similar in early training rounds before the accuracy for weaker privacy levels (i.e., smaller noise scale) becomes higher in the final training rounds. See, e.g., Figure 7 in (Kairouz et al., 2021a).
>
> 3) Practical Setting and Motivation:
>
> Differentially Private Federated Learning has already been widely deployed in practice (see Section 1). In many cases, there could be regulations and/or other privacy concerns that prevent even a central server from obtaining sensitive client information, or that prevent clients from trusting that the central server will properly add noise. This is the setting in which our results fit—clients must add noise locally. We believe our experiments show the practicality of our solution.
>
> Moreover, our setting is the same as many other works, including an ICML 2021 paper (Kairouz et al., 2021a), and a concurrent work (Ball et al., 2024). (See our comparison to the latter paper, which has weaker privacy guarantees, in Section 1, Related Work and Section E.)
>
> 4) “Theoretical analysis heavily depends on existing works”:
>
> We believe that our proposed Constant-Overhead Linear Secret Resharing protocol and its correctness/security analyses are novel and significant contributions.

---

### Official Review · Reviewer_p54v · 2025-03-13

**Overall Recommendation:** 3

**Summary:**

The paper focuses on differentially private (DP) federated learning (FL), with the main contribution of formulating a distributed version of the well-known matrix mechanism. As the continuous Gaussian mechanism typically used with the centralized matrix mechanism does not work well with cryptographic primitives, the proposed method is based on the discrete Gaussian mechanism.

The paper proposes a method with constant-overhead (in communication, compared to squared scaling of a naive approach) based on a suitable linear secret sharing protocol, and includes a formal proof that the construction  is secure. The proposed method is also shown to be robust against a fraction of malicious or failing parties.

The paper includes empirical experiments on some data sets and models (Stack Overflow with LSTM model, FEMNIST with small CNN) demonstrating that the proposed method outperforms a local variant of discrete Gaussian DP mechanism.

### Update after the rebuttal and discussions

I raise my score to weak accept reflecting the discussions. I am still not perfectly happy with the paper, e.g., the authors should use more recent noise mechanism to be able to claim being SOTA (while this was discussed, I do not accept this as a made change without seeing some evidence of the change actually being implemented), I am somewhat hesitant about the contribution (as discussed, as the main contribution is a cryptography protocol, I think this would be a more suitable submission in a more cryptography oriented venue to get more relevant estimation of the novelty and importance of the said protocol; beyond this concern however, I do think the paper is within scope for ICML), and generally the paper could be written more clearly.

**Claims And Evidence:**

Some of the claims are not properly supported by the evidence (see Questions for authors for more details):
* Claims about beating current SOTA, when the experiments only use discrete Gaussian mechanism also for the baselines.

**Essential References Not Discussed:**

The paper fails to mention several references, e.g., DP noise mechanisms improving on the discrete Gaussian (see Questions for authors for some concrete examples).

**Experimental Designs Or Analyses:**

I have checked all the reported empirical results, and have some doubts and questions on the implementation (see Questions for authors for details).

**Methods And Evaluation Criteria:**

The methods and evaluation mostly seem to be ok and can be used for the stated purpose. I find the complexity evaluation to be somewhat unclear (see Questions for authors for more details).

**Other Comments Or Suggestions:**

* Eg: lines 19-20, 1st col; 29-33, 2nd col etc.: please do not call your distributed matrix mechanism local DP (LDP), as LDP has a well-established meaning that does not correspond to the proposed setting (in LDP each client guarantees DP on their own data independently of others). Instead, distributed DP (DDP) is often used in settings where LDP-type noise is combined with cryptographic primitives such as SecAgg for enhanced privacy-utility trade-offs.
* I find the description of the LRP in Sec.3 to be somewhat confusing, maybe partially because of notations; e.g., looking at the Reshare, is the idea that the input is a list of $k$ shares, and the output then length $k$ list of shares, where each element is again a list of $k$ shares?

**Other Strengths And Weaknesses:**

### Strengths
* The problem of formulating a distributed matrix mechanism is an interesting and, I believe, an important one.
* The proposed method has several nice properties (constant overhead, fault-tolerance).
* The code is provided in the supplement.

### Weaknesses
* The empirical comparisons do not include state-of-the-art DDP baselines, even though such methods should be well-suited to the problem given the assumed trust model and the available secure primitives.
* Considering a typical FL setting, requiring client-to-client communications in cross-device setting between clients for each step is a heavy requirement.
* I find the writing to be somewhat confusing at times (although this might be partially due to not being an expert in cryptography).
* The main contribution of the paper is a cryptography protocol. While I do not think this is exactly out of scope for ICML, I do think that there might be other venues better suited for this kind of work.

**Questions For Authors:**

### Critical

1) Sec. 5.: Please clarify how exactly the baseline is run (e.g., how all the hypers such as clipping have been set, how is subsampling amplification calculated).

2) Lines 407-409: how does the number of rounds affect the complexity of the algorithm?

3) Lines 296-299: how is the scaling of the proposed method in terms of $k$?

4) Cf. lines 99-101, 2nd col: there have been several improved noise methods proposed after the discrete Gaussian (see, e.g., Agarwal et al. 2021: The Skellam Mechanism for  DP FL, Chen et al. 2022: The Poisson binomial mechanism...). Is there some specific reason to instead use the discrete Gaussian as the noise mechanism in the current paper and claim that it is state of the art? Note that there are also improved centralized MM papers not referenced, e.g., Choquette-Choo 2024: (Amplified) Banded Matrix Factorization: A unified approach...

5) Does the proposed method assume that each client always runs a single optimization step, or does it work with pseudo-gradients resulting from running multiple local steps?
### Non-critical
* Lines 269-270: should the number of non-faulting honest parties be $n-t_d-t_c$?
* Lines 252-254: so does this mean that in the naive scheme each party $i=1,\dots,n$ chosen at round $T$ needs to send $n$  shares to each of the $j=1,\dots,n$ parties chosen for round $T+1$?

**Relation To Broader Scientific Literature:**

There is some discussion relating the current paper to the more general field of distributed DP ML. I think this is good enough given the topic.

**Theoretical Claims:**

I did not have chance to check the correctness of proofs.

---

> ### Author Rebuttal · Authors · 2025-04-01
>
> 1) How baseline is run:
>
> We use the exact experiment details specified in the respective papers: (Section 5; Kairouz et al., 2021) for Local DP DDG and (Section 6 and Appendix I; Choquette-Choo et al., 2023a) for the Central DP mechanisms. For local DP DDG, the FEMNIST norm clip is set to 0.03 and the SO-NWP norm clip is set to 0.3. Additionally, Generic RDP amplification via sampling (Zhu And Wang, 2019) is used. For central DP, the SO-NWP clip is set to 1. Amplification is calculated using the technique presented in their paper along with the dp_accounting python library (reference [18] in their paper). We will include these in our paper.
>
> 2) Effect of number of rounds on complexity:
>
> Let T* be the total number of rounds and d the model dimension. In a given round T <= T*, the complexity of our algorithm for a general matrix factorization (including the Optimal Matrix Factorization) scales as O(d * T), while for the Honaker Online mechanism, the complexity scales as O(d log T). We will make this more explicit in the paper.
>
> 3) Scaling in terms of k:
>
> We use k = 2 * \mu * n for 0 < \mu < 1/2, as stated in Section 3, “Communication Complexity”. In round T, the concrete overhead is a 1/(4 \mu^2) factor (times d * T in the case of general, including optimal, Matrix Factorization, and d log T in the case of Honaker Online mechanism). We will make this clearer in the paper.
>
> 4) Noise methods and centralized MM papers:
>
> Since our techniques are independent of the noise distribution, we just chose one that we believe to be the most well-studied/understood: the discrete Gaussian. Indeed, we just need CX + Z to be differentially-private, where Z can be sampled from any (discrete) noise distribution; whichever distribution yields the best results can be used and these results will be at least as good as those in our paper. Ultimately, the message of our paper is the same: using the matrix mechanism for local (distributed) DP is better than using uncorrelated noise for local (distributed) DP.
>
> Also, we believe that the Skellam Distribution is incomparable to the discrete Gaussian in terms of privacy-utility tradeoff: Theoretically, quoting from (Agarwal et al., 2021) the bound on \eps DP they “provide is at most 1 + O(1/\mu) worse than the bound for the Gaussian” (this also holds with respect to the discrete Gaussian). Experimentally, it can be seen from Figure 5 of (Agarwal et al., 2021) that performance from the discrete Gaussian and Skellam are quite similar; depending on \eps, sometimes the discrete Gaussian is better.
>
> For the Poisson binomial mechanism of (Chen et al. 2022), Table 1 of that paper indicates that the error from that mechanism and the discrete Gaussian is asymptotically the same. They do not experimentally compare to the discrete Gaussian or run FL experiments, so it is hard to tell which has better accuracy in practice. We will be sure to cite this paper.
>
> We are happy to include the performance of the Skellam and Poisson binomial distributions in our paper (in the current version we do acknowledge and reference the Skellam Distribution in Footnote 6).
>
> Also, note that we do reference and indeed compare to “Choquette-Choo: (Amplified) Banded Matrix Factorization: A unified approach…” as the SOTA, which is indeed a NeurIPS 2023 paper and thus cited as (Choquette-Choo et al., 2023) in e.g., caption of Figure 2, and elsewhere.
>
> 5) Pseudo-gradients:
>
> Our system only requires clients to compute vectors that will be used to update the model parameters at the server (by somehow aggregating all of the vectors of the clients, i.e., averaging). So pseudo-gradients can be used in our system.
>
> 6) Lines 269-270:
>
> Yes, thank you for pointing it out.
>
> 7) Lines 252-254:
>
> Yes, each party i = 1,...,n in round T sends n shares, for a total of n^2 communication per round across all i=1,...,n.
>
> 8) Client-to-client communication:
>
> Note: in practice the communication between clients will be (end-to-end encrypted and) routed through the server. Indeed, it is quite common for communication to be routed through the server, see, e.g., (Bonawitz et al, 2017) and most follow-ups. Moreover, such communication only occurs between clients of consecutive cohorts (i.e. rounds T and T+1), so we do not believe it's a heavy requirement.
>
> 9) Suitability of ICML:
>
> We believe that any paper that improves the SOTA for any topics of interest in the Call for Papers should be seen as suitable. Our paper is for “Trustworthy Machine Learning” topic listed in the call. Indeed, we directly improve an IMCL 2021 paper (Kairouz et al. 2021a).
>
> 10) LDP -> DDP:
>
> Thanks. We will fix it.
>
> 11) LRP description:
>
> The input to Reshare is indeed a vector of k shares: z_[1,k]^i = (z_1^i, …, z_k^i). The output is a packed secret sharing where the shared secret vector is the above, z_[1,k]^i. This output sharing is denoted by [z_[1,k]^i] = ( (z_[1,k]^i)^1 ,... , (z_[1,k]^i)^n) (i.e., a vector of n shares, 1 for each party in the next cohort).

---

> > ### Comment · Reviewer_p54v · 2025-04-05
> >
> > Thanks for the clarifications, I will raise my score accordingly. Here are some final notes:
> >
> > 4. I do think that using the SOTA discrete noise is better than using the discrete Gaussian, while I also agree that depending, e.g., on the actual privacy budget used or the communication budget, the observed difference in FL performance might be small. At the very least, you should not claim that discrete Gaussian is the current SOTA.
> >
> > 9. As I said, I do not claim that the work is out of scope for ICML. However, as the ML part otherwise seems very standard DP FL, the most important contribution is the cryptography part (as well as the specific application to DP FL of course). Obviously,  judging the novelty and importance of the cryptography contribution is easier for reviewers familiar with the field.

---

> > > ### Author Response · Authors · 2025-04-07
> > >
> > > Thank you for kindly considering our responses to your concerns. We are more than happy to include experiments/comparisons for the Skellam and Poisson binomial distributions, and change textual references to the “SOTA discrete noise distributions” in the final version of the paper. Please let us know if there are any remaining concerns that you would like us to address before you formally increase your score.

---

### Official Review · Reviewer_QYjM · 2025-03-14

**Overall Recommendation:** 3

**Summary:**

This paper presents a protocol for long running secure aggregation with matrix mechanism DP. The protocol consists of a sequence of committees/data-providers passing theinternal secrets of the algorithm along in secret shared form. The main theoretical innoviation is the novel secret resharing scheme used to pass large amounts of secret shared data along in a linear and communication efficient manner.

**Claims And Evidence:**

The support for the claims is presented though I have a couple of concerns which I address in the next boxes.

**Essential References Not Discussed:**

None I am aware of.

**Experimental Designs Or Analyses:**

I didn't check the details of the experiments

**Methods And Evaluation Criteria:**

The assessment in the experiments section does make sense. It would have been nice to see a comparison to the optimal matrix mechanism in the clear i.e. without discretization but this would merely be a nice to have.

**Other Comments Or Suggestions:**

None.

**Other Strengths And Weaknesses:**

One mistake I think should be pointed out is in the description of Ball et al. This paper states that paper suffers from the server being able to submit the wrong ciphertexts for decryption. That is a misconception, in that paper the information the clients use for decryption is the secret key they have shared amongst them and the public parameters used for encryption neither of which come from the server (except in the sense that the initial seed for generating the public parameters may come from the server). Having said that this paper can be justified alongside Ball et al by the novelty of its approach so the valitiy of this paper shouldn't depend on them having that privacy advantage over Ball et al.

If their protocol is not resistant to any dropouts (which it isn't as written) that seems a major weakness for any hope of a practical implementation. Fixing this is the main substantive thing stopping my review from being more enthusiastic.

**Questions For Authors:**

Is your protocol supposed to be dropout resistant? If so how would you deal with the problem I highlighted about co-ordination for Recover?

Have I misinterpreted Ball et al? If not can you fix that description for the camera ready?

**Relation To Broader Scientific Literature:**

The Matrix mechanism work in the central model is a big new thing. Putting that in a secure distributed setting is an obvious and potentially impactful expansion combination that makes a lot of sense. The only other paper attempting to do this I am aware of is the Ball et al pre-print. Ball et al uses very different methods (in a similar threat model) so this is a good additional angle of attack and the ideas might well be combinable.

**Theoretical Claims:**

I verified the correctness and security of the secret sharing scheme. The paper claimed that Recon is a function of i and z_i and adding any t_c+k of the Recons would give the result. This is close tot he truth but not right because the \lambda_i depend on exactly which subset of clients are involved in the reconstruction. This led me to doubt whether the Recover function worked. It can be made to work but only if the client in cohort T+1 agree on a subset of the clients from cohort T to use information from in the reconstruction (whom they have all received shares from). This would mean they need to know information about which messages get through to everyone, this is probably achieveable without too much expense (maybe if the communication all goes via the server it could play a co-ordinating role) but it should be specified how this is going to be achieved. I suspect the authors missed this was necessary because they oversimplified the description of how Recon works. Alternatively maybe they mean for their protocol to not be resistant to any lost messages or dropouts and thus the set being used should always be all n=t_c+k clients.

---

> ### Author Rebuttal · Authors · 2025-04-01
>
> 1) Dropout Resistance:
>
> We thank the reviewer for pointing this out. We did not include the details for dropout resistance in the main body to make the (already complex) protocol easier to understand, but we should have included it in the appendix.
>
> However, the reviewer is correct; there is a simple way to fix this: The clients in cohort T+1 must agree on a subset of clients from cohort T whose shares they use in the Recover algorithm. As the reviewer suggests, the server (through which all communication is routed) can simply send to cohort T+1 this subset directly, defined as those clients of cohort T who sent shares to *all* cohort T+1 clients. (For reference, it is quite common for communication to be routed through the server, see, e.g., (Bonawitz et al, 2017)).
>
> Then, the Recover algorithm of protocol LRP in Section 3 will additionally take as input such a set of NONDROPOUTS (received from the server), and the existing summation in the Recover algorithm will now be for i \in NONDROPOUTS (and similarly for the Recons algorithm).
>
> We will make this change for dropout resistance in the final version.
>
> 2) Description of Ball et al:
>
> Note that in Ball et al, the server receives ciphertexts from the clients, then maintains and aggregates the ciphertexts which are to be eventually decrypted (also by the server). For example, in Section 4.1 of that paper, it is written that from an encryption of x1 received from Alice and x2 received from Bob, “the server can sum the result to get an encryption of x1+x2”. Then “imagine the server is holding a ciphertext” encrypting x, “Alice and Bob can simply compute and send” their (noisy) secret keys which “enables the server to recover x”.
>
> Also, in Section 4.2, it is written that “writing to the secret state” means “the server then simply sums the resulting correlated ciphertext with all ciphertexts received from cohort Ci”.
>
> Thus, the server maintains the ciphertexts itself and even performs the decryption itself, so it can certainly submit the wrong ciphertext for decryption. Furthermore, even if the server did not perform the decryption itself, the clients would still need to receive the maintained (aggregated) ciphertexts from the server.
>
> 3) Comparison to matrix mechanism in the clear (without discretization):
>
> Note that we indeed compare to the matrix mechanism in the clear (without discretization) for the StackOverflow NWP task in FIgure 2. We achieve very similar accuracy for given privacy values \eps.

---

> > ### Comment · Reviewer_QYjM · 2025-04-07
> >
> > I am happy with the response regarding dropouts, I'm sure the authors can make that change fine for the camera ready.
> >
> > In Section 4.1 of Ball et al The quote you give  of "Alice and Bob can simply compute and send" continues with noisy As not noisy s, which depends on A. A is a publicly known parameter used in making the ciphertext, the client know this prom the specification of public parameters for the computation and do not have to be informed of it by the server. It is true that only the server ever holds the ciphertext and decrypts it using the noised expanded keys from the clients, but the clients choose which A to use and in so doing guarantee that the As+e they are sending in can only be used to decrypt the ciphertext that is supposed to be decrypted. If what the client's were sending in wasn't specific to what was supposed to be decrypted the protocol wouldn't even be semi-honestly secure. The attack you are suggesting doesn't work.
> >
> > Regarding 3, my apologies for missing that figure, this seems fine.

---

> > > ### Author Response · Authors · 2025-04-07
> > >
> > > Thank you for the kind responses regarding dropouts and Figure 2.
> > >
> > > Let us provide the details of one of the attacks on Ball et al. that we mention in Section 1, Related Work and Appendix E of our paper:
> > >
> > > The encryption scheme in Section 4.1 of Ball et al. works with the server holding the ciphertext As + Te + x (where s = s_1 + s_2). For decryption, Alice and Bob send the server As_1 - Te_1 and As_2 - Te_2, respectively, so that the server can decrypt (As + Te + x) - (As_1 - Te_1) - (As_2 - Te_2) = x + T(e + e_1 + e_2) \equiv_T x. (Note that \equiv_T is the notation that Ball et al. use for computing a value (such as x + T(e + e_1 + e_2)) mod T, where T is a public parameter of the encryption scheme).
> > >
> > > Our attack works by the server first multiplying the ciphertext by 100 to get 100 * (As + Te + x). Alice and Bob, believing that the server wants to decrypt the correct ciphertext, will send to the server the same As_1 - Te_1 and As_2 - Te_2, respectively, as above. Then, the server can compute 100 * (As_1 - Te_1) and 100 * (As_2 - Te_2) and finally decrypt 100 * (As + Te + x) - 100 * (As_1 - Te_1) - 100 * (As_2 - Te_2) = 100 x + 100 * T (e + e_1 + e_2) \equiv_T 100 x.
> > >
> > > Thus, the server can obtain a ciphertext encrypting 100x instead of x by just performing local operations.
> > >
> > > In the context of DP FL with the matrix mechanism (their Section 3.3), this ruins the DP guarantees of the mechanism CX + Z (Or CX + \eta in their notation). Indeed, the clients “store the inputs in the odd numbered rounds”, i.e., just send encryptions of their gradients x without noise to the server, “and then in each even numbered round reveal a new instance of noise with the appropriate linear combinations of the inputs on top.” Thus, gradients are first encrypted without noise. Using the above attack, the server can then locally compute a ciphertext encrypting 100x instead of x (the actual gradient(s)), and since the noise scale is tuned to the norm of x, and not 100x, the DP guarantee will be violated.
> > >
> > > We will clarify this attack in the final version of the paper. Let us know if there are any other remaining concerns.

---

### Decision · Program_Chairs · 2025-05-01

**Decision:**

Accept (poster)

**Comment:**

This submission proposes a distributed matrix mechanism (DMM) for federated learning (FL) with local differential privacy (DP), aiming to achieve a better privacy-utility tradeoff using a novel cryptographic protocol for secure value transfer with constant communication overhead.
While the idea of combining the matrix mechanism with cryptographic techniques in a distributed setting is interesting and potentially impactful, several concerns raised by the reviewers, particularly regarding the setting and assumptions, the experimental evaluation and the comparison to the prior work (in particular the state-of-the-art differentially private federated learning baselines), as well as the writing and clarity of exposition. It would be preferable if the paper addresses these concerns before getting published at a venue such as ICML.